# Structural basis for the inhibition of IAPP fibril formation by the co-chaperonin prefoldin

Ricarda Törner [1,5], Tatsiana Kupreichyk [2,3,5], Lothar Gremer [2,3,4], Elisa Colas Debled [1], Daphna Fenel[1], Sarah Schemmert[2], Pierre Gans[1], Dieter Willbold [2,3,4], Guy Schoehn [1], Wolfgang Hoyer [2,3 ✉] & Jerome Boisbouvier [1 ✉]

Chaperones, as modulators of protein conformational states, are key cellular actors to prevent the accumulation of fibrillar aggregates. Here, we integrated kinetic investigations with structural studies to elucidate how the ubiquitous co-chaperonin prefoldin inhibits diabetes associated islet amyloid polypeptide (IAPP) fibril formation. We demonstrated that both human and archaeal prefoldin interfere similarly with the IAPP fibril elongation and secondary nucleation pathways. Using archaeal prefoldin model, we combined nuclear magnetic resonance spectroscopy with electron microscopy to establish that the inhibition of fibril formation is mediated by the binding of prefoldin's coiled-coil helices to the flexible IAPP N-terminal segment accessible on the fibril surface and fibril ends. Atomic force microscopy demonstrates that binding of prefoldin to IAPP leads to the formation of lower amounts of aggregates, composed of shorter fibrils, clustered together. Linking structural models with observed fibrillation inhibition processes opens perspectives for understanding the interference between natural chaperones and formation of disease-associated amyloids.

[1] University Grenoble Alpes, CNRS, CEA, Institut de Biologie Structurale (IBS), 71, Avenue des Martyrs, F-38044 Grenoble, France. [2] Institute of Biological Information Processing (IBI-7: Structural Biochemistry) and JuStruct: Jülich Center for Structural Biology, Forschungszentrum Jülich, 52425 Jülich, Germany. [3] Institut für Physikalische Biologie, Heinrich-Heine-Universität Düsseldorf, 40225 Düsseldorf, Germany. [4] Research Center for Molecular Mechanisms of Aging and Age-Related Diseases, Moscow Institute of Physics and Technology (State University), Dolgoprudny, Russia. [5] These authors contributed equally: Ricarda Törner, Tatsiana Kupreichyk. ✉email: wolfgang.hoyer@hhu.de; jerome.boisbouvier@ibs.fr

To date there are about 50 proteins or peptides identified which are implicated in amyloid diseases[1]. The hallmark of these pathologies is the formation of thermodynamically highly stable fibrils with a characteristic β-cross structure[1,2] from native monomeric proteins. Most studied amyloidogenic proteins are involved in neurodegenerative diseases such as Alzheimer's or Parkinson's disease[3]. However, the cell-toxic formation of fibrillar aggregates can also be of systemic nature or localized in peripheral organs and can be associated with very different diseases such as arthritis or diabetes[4]. The 37 amino-acids long intrinsically disordered peptide hormone islet amyloid polypeptide (IAPP)[5], also known as amylin, is found as amyloid aggregates surrounding β-cells in the pancreas in 90% of type II diabetes cases[6,7]. A variety of ways have been identified how amyloids can confer toxic activities, for example the increase of the permeability of cell membranes by oligomers of amyloidogenic proteins[8], and formation of porous structures[9]. Likewise, in the case of IAPP, not mature fibrils but small oligomeric species formed as intermediates during the fibrillation process have been observed to be the most toxic for β-cells[10]. Nevertheless, the mature amyloid fibrils, whose structures have recently been elucidated for IAPP[11–15] are not innocuous as they can sequester proteins of the cell machinery[16,17], exert mechanical stress on the cell[1], and play a significant catalytic role during the fibril formation process[18].

The fibrillation process starts with an oligomerization event, where a nucleus is formed, which rapidly elongates by templated incorporation of monomers. As soon as nucleation has occurred, the fibril growth proceeds exponentially since not only elongation occurs, but also secondary processes, such as fibril fragmentation or secondary nucleation on fibril surfaces, which increase the rate of the fibrillation process[19]. In vivo, under normal conditions, the protein quality control machinery composed of chaperone systems, ubiquitin-proteasome systems, and autophagy-lysosomal systems is able to prevent the formation of aggregates[20,21]. However, in cases of misfolding diseases, these systems are either weakened due to a decrease of capacity during aging[16] or overwhelmed due to increased concentrations or more-aggregation prone mutants of amyloidogenic proteins in hereditary conditions[4]. A detailed mechanistic understanding of this natural quality control machinery allows for targeting its processes for medical intervention, either by targeted upregulation of its components or via functional replacement with small molecules[22]. Chaperone molecules as modulators of protein conformational states are key factors acting on proteins during their transition from native state to the amyloid fold[20,23,24]. Interacting with a multitude of species, chaperones are able to alter the fibrillation process by inhibition of aggregation, disaggregation, or detoxification[25–30]. For instance, fibrillation inhibition can be mediated via interaction with misfolded monomers[31] or small oligomers, which prevents the formation of seeding competent nuclei, but also interaction with fibril ends[32] or decoration of the fibril surface[33] is possible, which inhibits elongation and secondary nucleation processes respectively. Treatment of the diseases associated with amyloidogenic proteins whose native state is intrinsically disordered remains an important challenge for modern medicine, so a better understanding of the modulation of such amyloidogenic proteins by chaperones is of paramount interest.

Here we report the mechanistic study of IAPP fibrillation inhibition by the HSP60 type II co-chaperonin prefoldin (PFD). Homologs of this co-chaperonin are found in the cytosol of archaeal and eukaryotic cells. The heterohexameric PFD is a holdase with a characteristic jellyfish architecture, consisting of a β-barrel body and coiled-coil α-helix tentacles[34]. Prefoldin binds substrates via a clamp-like mechanism and delivers it to HSP60

for refolding[35]. PFD has previously been shown to interact with different disease-relevant amyloidogenic substrates and to inhibit their aggregation[36–39], however no structural or mechanistic insights in this process have been described heretofore. We have integrated kinetic investigations with structural studies using atomic force microscopy (AFM), electron microscopy (EM) and nuclear magnetic resonance (NMR) spectroscopy to elucidate the different inhibition pathways and to provide a structural understanding of the prefoldin-amyloid interaction. The study of the highly dynamic complex between the 90 kDa PFD and intrinsically disordered 4 kDa IAPP enabled us to obtain insights into the chaperoning process at sub-molecular level. We show that inhibition of elongation and secondary nucleation is achieved by the interaction of PFD with IAPP fibril ends and surface, which subsequently leads to lower fibrillation rates and to the formation of clustered aggregates instead of individual fibrils.

## Results

**PFD impedes IAPP aggregation**. IAPP aggregation kinetics in presence of PFD show a strong inhibitory effect of PFD on IAPP fibril formation in vitro (Fig. 1a and Supplementary Fig. 1a). For both *Pyrococcus horikoshii* and human PFD (denoted as PhPFD and hPFD, respectively), the overall tendencies are highly similar. The aggregation process is affected at low substoichiometric PFD concentrations, suggesting an interaction between PFD and IAPP oligomers or fibril states, rather than a simple sequestration of IAPP monomers by PFD. In de novo aggregation assays monitored by thioflavin T (ThT) fluorescence, starting from monomeric IAPP without addition of any pre-formed aggregates, progressive addition of PFD does not have a major effect on the duration of the lag phase (the initial phase during which no aggregation is detectable by ThT fluorescence) but strongly reduces the fluorescence intensity (FI) in the final plateau phase (Fig. 1a, top row, and Supplementary Fig. 1a). In addition, increased PFD concentrations result in a significant reduction of fibril growth rates associated with increased durations of the growth phase (the time span between lag phase and plateau phase) by more than an order of magnitude (Fig. 1a, top row, and Supplementary Fig. 1a).

We tested the effects of human and archaeal PFD on two specific steps of IAPP fibril formation, fibril surface-catalyzed secondary nucleation and fibril elongation, by performing aggregation assays under conditions where one of these steps is dominating the overall aggregation kinetics. Secondary nucleation has been identified as a critical step in IAPP fibril formation, and has been linked to the emergence of toxic IAPP aggregates[18,40]. Under conditions with an active secondary nucleation pathway, addition of fibril seeds bypasses the need for primary nucleation[41]. We therefore tested the effect of PFD on IAPP assembly in the presence of preformed IAPP fibril seeds (Fig. 1a, middle row). In order to suppress aggregation pathways that are independent of secondary nucleation: 1) we did not agitate the samples, as agitation promotes the amplification of aggregates formed by primary nucleation, by increasing the effective size of surfaces involved in primary nucleation, by enhancing mass transport, and by promoting mechanical fibril breakage[42], and 2) we did not sonicate the preformed fibrils, which would promote aggregation by pure elongation of the seeds. Under these conditions of dominant secondary nucleation, maximal growth rate drastically decreases for [PFD]:[IAPP] molar ratios above 1:113, and drops to almost zero for molecular ratios larger than 1:17, indicating that PFD effectively interferes with the fibril surface-catalyzed generation of new fibril nuclei. Substoichiometric PFD concentrations suffice, suggesting that fibril surfaces are the target sites of PFD in inhibition of secondary nucleation

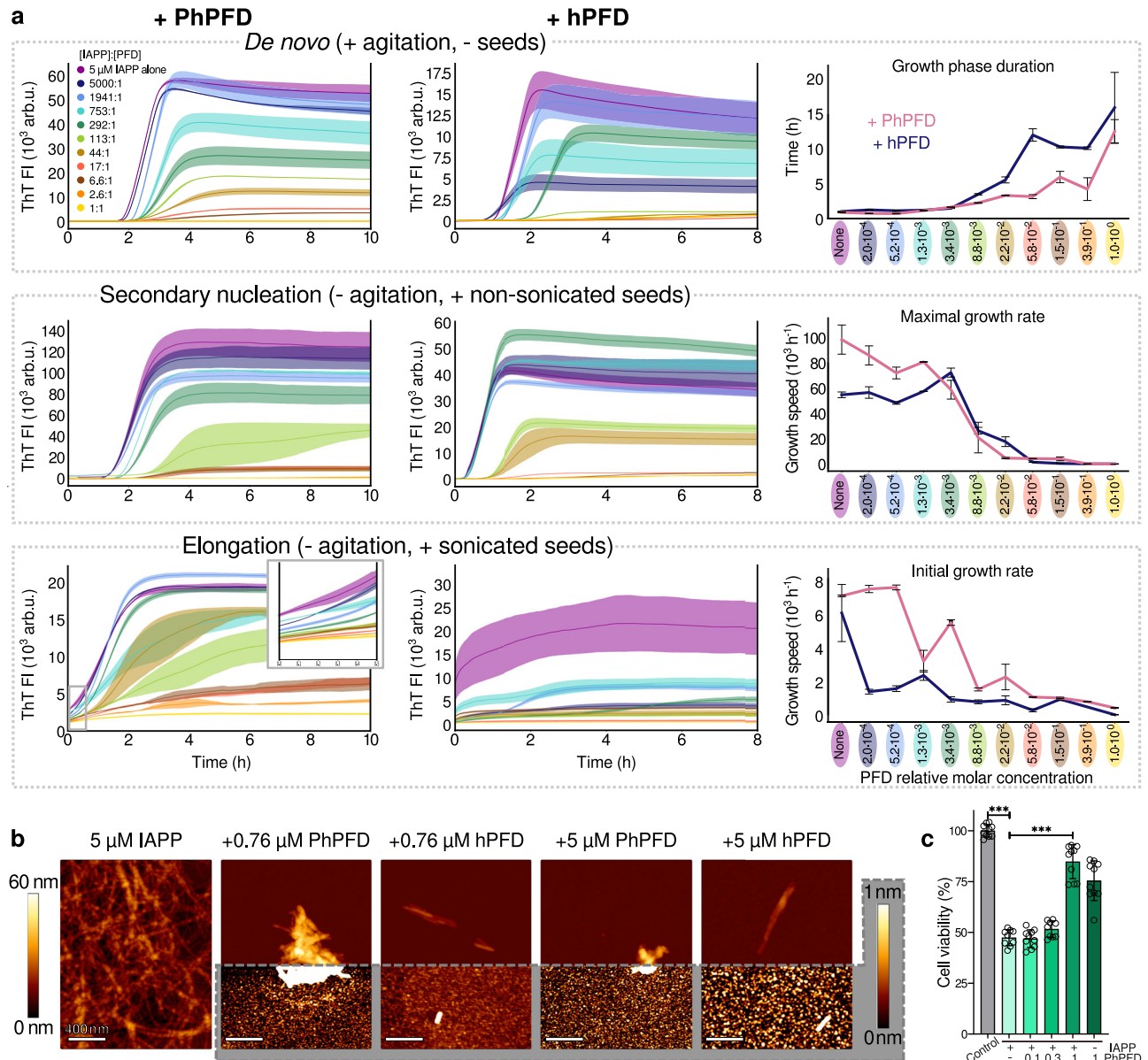

**Fig. 1 Inhibition of IAPP aggregation by prefoldin. a** IAPP (5 µM) aggregation kinetics in absence and presence of PhPFD (left column) and hPFD (middle column) with [PFD]:[IAPP] molar ratios ranging from 1:5000 to 1:1, followed by ThT fluorescence intensity (FI). Top row: de novo aggregation assay, starting from monomeric IAPP only, in presence of agitation. Middle row: secondary nucleation assay, starting from a mixture of monomeric IAPP and non-sonicated preformed IAPP fibrils (seeds), without agitation. Bottom row: elongation assay, showing non-agitated aggregation of monomeric IAPP in presence of shorter sonicated seeds, where the initial linear parts of the curves (inset) are reflecting pure elongation of fibril seeds. The right column shows extracted kinetic parameters: growth phase duration, maximal growth rate, and initial growth rate. Further kinetic analysis is presented in Supplementary Fig. 1a. Every curve represents a mean of $n = 3$ replicates with the error shadows illustrating SD; the kinetic parameters are shown as mean ± SD. **b** AFM images of the samples at the end of the aggregation assays in section a (top row). Scale bar corresponds to 400 nm. In order to visualize objects decidedly differing in height, two color code scales were used: the gradient from dark brown to white represents either 0–60 nm height, or 0–1 nm (highlighted in gray). The full AFM overview is presented in Supplementary Fig. 2, including the control sample containing PFD alone and illustrating that the carpet-like coverage of the surface with small roundish particles at increased PFD concentration is the appearance of PFD itself. **c** PhPFD reduces the IAPP-induced negative impact on cell viability of RIN-m5F cells. After incubation of RIN-m5F cells with 1 µM IAPP w/o or with several concentrations of PhPFD (0.1, 0.3, 1 µM), an MTT cell viability test was conducted. Data is represented as mean ± SD (out of n=2 independent experiments with four to five technical replicates), one-way ANOVA with Tuckey post hoc analysis, ***$p \leq 0.001$.

(see Arosio et al. for an overview of potential target species for inhibition of specific steps of amyloid fibril formation[43]).

Fibril elongation, i.e., the binding of monomers to fibril ends and their conformational conversion into the fibrils' cross-β structure, is dominating the aggregation kinetics when a high concentration of fibril ends is present[44]. For observation of PFD's effect on elongation of IAPP fibrils, we therefore added sonicated,

short, pre-formed IAPP fibrils to offer a high number of fibril ends (Fig. 1a, bottom row). Under these conditions, the very first phase of linear growth reflects the pure elongation process, while the subsequent exponential increase indicates that secondary nucleation starts to contribute. Addition of PFD results in a distinct decrease of the initial growth rate, demonstrating that PFD interferes also with fibril elongation. Substoichiometric PFD

concentrations are sufficient, indicating that fibril ends are the target sites of PFD in inhibition of fibril elongation.

While the described effects on IAPP fibrillation are present for both archaeal and human PFD, the inhibitory effects of hPFD are more pronounced. The kinetic study of different modes of IAPP aggregation in presence of PFD shows that PhPFD and hPFD affect various steps of the fibrillation process. Moreover, the final plateau values in all the kinetic experiments become significantly lowered if PFD is present (Fig. 1a).

**PFD induces formation of compact aggregates**. Atomic force microscopy (AFM) of the IAPP species present after the aggregation kinetics experiment showed changes in the appearance and the amount of IAPP aggregates in the presence of PFD (Fig. 1b and Supplementary Fig. 2a, b). IAPP aggregation in absence of PFD results in a large amount of long, individually distinguishable fibrils (Fig. 1b left). In comparison, IAPP aggregates in presence of PFD are rare and rather clustered. The increase of PFD concentration lowers the overall amount of IAPP aggregates. Although the clustering gives rise to bigger assemblies of IAPP, some single fibrils are clearly visible at their perimeters (marked with arrows in Supplementary Fig. 2). Hence, this suggests that the observed clusters are not amorphous, but they rather represent bundles of shorter IAPP fibrils.

**PFD increases the viability of rat pancreatic beta cells exposed to IAPP aggregates**. To assess the effect of PFD on IAPP toxicity, we evaluated the viability of rat pancreatic RIN-m5F cells upon addition of IAPP with and without PFD in an MTT (3-(4,5-Dimethylthiazol-2-yl)-2,5-diphenyl-tetrazolium bromide) assay. When monomeric IAPP at a concentration of 51 μM was incubated for 3 days and diluted into the cell culture medium to a final concentration of 1 μM, cell viability reduced to 47.4% of that of the non-treated control (Fig. 1c). Co-incubation with PhPFD at molar ratios between 1:10 and 1:1 showed a concentration-dependent rescue of cell viability, with cell viability reaching 84.8% of the control at equimolar ratio of PFD. This demonstrates that PFD inhibits IAPP-induced cell toxicity (Fig. 1c).

**IAPP binds PFD cavity in a dynamic manner**. In order to determine the binding site of IAPP on the PFD surface, NMR binding studies were performed with PhPFD, whose NMR assignment of backbone and methyl groups is available[45] (Supplementary Fig. 3). Structurally, hPFD and PhPFD are similar[46,47], but whilst hPFD is composed of six different subunits, the PhPFD complex contains 2 α- and 4 β-subunits, which simplifies the investigation of this 86.4 kDa protein heterohexamer by solution NMR. The similar results obtained by AFM and ThT assays for the investigation of the effects of hPFD and PhPFD on IAPP fibril formation support the choice of this hyperthermophilic model system, which was used for all subsequent structural studies.

For the NMR investigation of the PhPFD-IAPP interaction, we have prepared highly deuterated complexes of PhPFD, selectively labeled on backbone ($^{15}$N-$^{1}$H probes) or end of side chains ($^{13}$CH$_3$ probes) on either α- or β-subunit. To determine the binding surface of IAPP on PhPFD, we prepared a spin-labeled IAPP construct, which allowed us to map the binding interface via paramagnetic relaxation enhancement (PRE)[48–50]. The disulfide bridge between cysteines two and seven on IAPP forbids spin labeling using standard thiol-reactive nitroxy derivatives, therefore, we linked a DOTA-cycle to the N-terminus using a two-β-alanine linker. The DOTA-functionalized IAPP was loaded with paramagnetic Gd$^{3+}$ (Gd-IAPP) or diamagnetic Lu$^{3+}$ cations (Lu-IAPP) (Supplementary Fig. 4a, b). Labeled PhPFD was incubated at 30 °C with either paramagnetic Gd-IAPP or diamagnetic Lu-IAPP reference (Fig. 2a–e). The residues involved in binding with IAPP were identified by comparing the intensities of PFD's NMR signals in presence of Gd-IAPP or Lu-IAPP, respectively (Supplementary Fig. 5). Mapping of the paramagnetically-broadened residues on the surface of PhPFD (PDB: 2ZDI, Fig. 2f) allows to locate the binding interface between IAPP and PhPFD (Fig. 2g–j). Whilst no PRE-broadening is observed for residues which are located on the top of the β-barrel body, broadening inside the cavity and along the sides of coiled-coil helices is detected. More specifically, strong interaction is observed in the middle part of the coiled-coil helices corresponding to residues 8-31 and 118-135 of the α-subunit. On the β-subunits the first 32 N-terminal residues are the most strongly broadened by the Gd-IAPP. These two regions face each other in the complex. All coiled-coil regions are especially enriched with glutamine. Interestingly, the regions strongly affected by IAPP paramagnetic relaxation present a globally negatively charged surface (−2) which can complement the net positive charge (+2) of the IAPP sequence. Also analysis of PFD's electrostatic surface indicates that the regions strongly affected by IAPP paramagnetic relaxation are predominantly negatively charged or neutral (Supplementary Fig. S6), suggesting that the electrostatic interaction may complement hydrophobic interactions, which presumably remain the main driving force for PFD-IAPP interactions. The large binding interface determined from the PRE-data suggests avidity-based binding to many lower affinity sites, which gives rise to the observed affinity. The NMR study suggests that monomeric IAPP is being incorporated in the chaperone cavity whilst binding and releasing the available interaction surfaces in a dynamic manner, as already observed for complexes involving unstructured proteins bound to other molecular chaperones[48,51–53].

**IAPP binds PFD with two binding segments**. To determine IAPP residues involved in binding of PhPFD, we recorded 2D NMR $^{15}$N-$^{1}$H-correlation spectra of U-[$^{15}$N]-labeled IAPP at 30 °C after addition of increasing ratios of PFD. Quantifications of the IAPP backbone amide signal broadening and chemical shift perturbations (CSPs) in the NMR titration experiment were used to map the binding sequence (Fig. 3a–d). Strong CSPs were observed for IAPP residues 10–19 and 24–28 (Fig. 3e). Peak broadening, resulting from apparent decreased tumbling rates upon binding of the 86.4 kDa prefoldin complex to small flexible IAPP, was located towards the N-terminus (Fig. 3d). C-terminal residues 34–37 are almost not impacted by interaction with PFD and are therefore most likely not involved in direct binding. Interestingly, binding is not restricted to the region 20–29, which is strongly involved in fibril formation and found buried in recently elucidated cryo-EM structures[11–15]. The spectral distribution in the proton dimension remains narrow, which suggests that IAPP does not fold upon binding, but remains essentially disordered. This confirms the PRE-results obtained upon monitoring the interaction of paramagnetically tagged IAPP on PFD which show an extended binding surface (Fig. 2). More quantitative analysis of binding strength was performed with the TITAN lineshape fitting program[54], assuming a 1:1 stoichiometry. Line-shapes and change of signal frequencies were fitted for Ala13, Val17, His18, Ser19, Leu27, Phe23, Ala25, and Ser28, as these residues are important for the binding and analysis is not hampered by peak overlap. The apparent $k_{off}$ rate was estimated by line shape fitting to be in the order of 6000 s$^{-1}$, corresponding to a residence time of IAPP on PFD in the order of 100–200 μs. The apparent dissociation constant $K_D$ (61 ± 3.2 μM - error determined by bootstrap analysis) shows that IAPP has an affinity for PFD in the low to medium range. Estimation of the dissociation constant $K_D$ by biolayer interferometry (BLI), for which monomeric biotinylated IAPP was

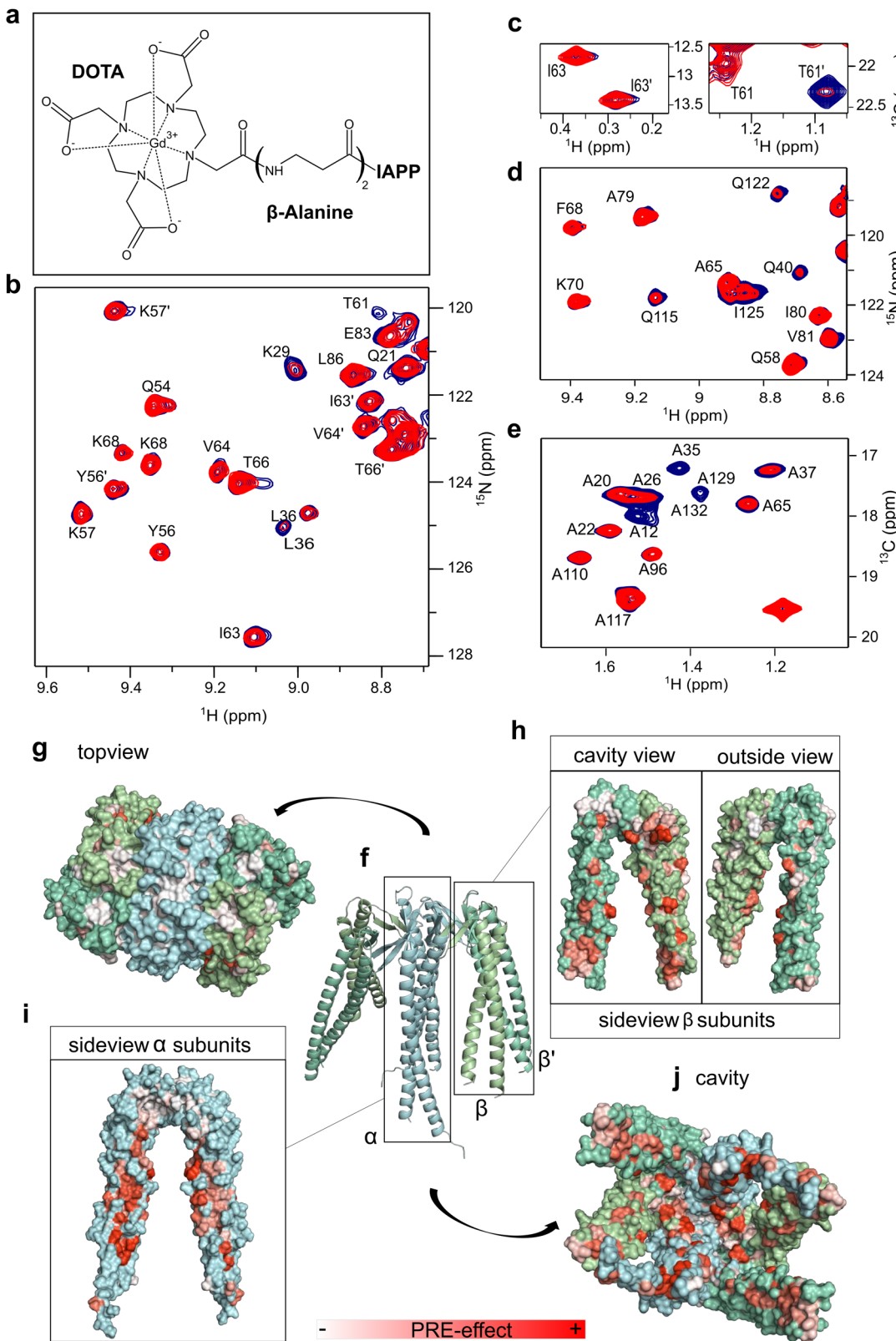

immobilized on a streptavidin sensor, gave a result in the same order of magnitude ($11 \pm 1.2\,\mu M$) (Fig. 3f).

**Inhibition is mediated by PFD binding to IAPP fibril ends and surface**. To understand the observed effect of substoichiometric inhibition of elongation and secondary nucleation observed in the

fibrillation assays, we have investigated PFD binding to IAPP fibrils by electron microscopy (EM). The major polymorph of IAPP fibrils used for this study has a pitch of ~48 nm (Fig. 4a–c), consistent with previous cryo-electron microscopy (cryo-EM) structure determination of IAPP fibrils (polymorph 1 in Röder et al. 2020[13], Fig. 4d). Preformed IAPP fibrils were incubated with PFD for 30 min, transferred to a carbon grid, stained with SST

**Fig. 2 Binding sites of IAPP on PhPFD structures. a** PRE-labeled IAPP was obtained by fusion of a DOTA-macrocycle to the N-terminus of IAPP and subsequent loading of DOTA with $Lu^{3+}$ (diamagnetic control) or $Gd^{3+}$ (paramagnetic). Line broadening resulting from the proximity of Gd-IAPP (red spectra), compared to the control Lu-IAPP (blue spectra), is visible in 2D $^{15}N$-TROSY spectrum of β-subunit (**b**), 2D $^{13}CH_3$-TROSY spectrum of β-subunit (**c**), 2D $^{15}N$-TROSY spectrum of the α-subunit (**d**), and $^{13}CH_3$-TROSY spectrum of the α-subunit of PFD (**e**). A partial peak doubling is observed for the β subunits, due to their position in the PFD heterohexameric complex, some of which could be assigned to β and β'[45] (panels **b**, **c**), while the two α-subunits are spectroscopically equivalent (panels **d** and **e**). **f** Model of PhPFD (PDB: 2ZDI) with α-subunits in blue, β-subunits in green. **g** No line-broadening was observed for residues on the top of the PFD complex. Conversely, a strong effect was observed inside the cavity (**j**) and along the elongated coiled-coil helices of (**h**) β/β'-subunits (two views related by 180° rotation) and (**i**) α-subunits dimer (side view).

and then visualized by electron microscopy. In the electron micrographs, strong decoration of the fibril with PFD is observed (Fig. 4e–h). Visual analysis shows that on the fibril surface approximately every 80 nm one PFD molecule is found, that is about 10% of the surface (Fig. 4e). Upon investigation of the fibril ends, a density was repeatedly observed towards the end of the fibrils (Fig. 4h), which suggests a binding of PFD also to the fibril termini – about 40% of ends are accompanied by a density.

To characterize this interaction further at a sub-molecular level, we have docked PFD (PDB: 2ZDI) on the previously determined cryo-EM structure of IAPP fibrils, based on experimental PRE-broadening on PFD and chemical shift perturbation detected by solution NMR between PFD and IAPP (Fig. 5d, e), using the software HADDOCK[55,56]. The docking procedure was either directed to the fibril surface or to the fibril ends, thereby creating two representative complex-models (Fig. 5d, e). Analysis of these models indicates that IAPP-PFD binding is strongly mediated by the N-terminal part of IAPP, which remains flexible and solution-exposed also in the fibrillar state, and that protruding N-termini of several neighboring IAPP molecules can interact with the same prefoldin (up to six). This model rationalizes the pattern observed in the negative stain micrographs, as well as the inhibitory effect of PFD on IAPP fibrillation in the kinetic assays.

## Discussion

Previously, the co-chaperonin prefoldin was reported to inhibit the fibrillation of highly toxic amyloidogenic proteins in vivo and in vitro[36–39], but no mechanistic or structural information was obtained so far. Here we apply IAPP as a model amyloidogenic substrate and show that prefoldin inhibits IAPP fibrillation at substoichiometric concentrations and interacts with multiple IAPP species, such as monomers and mature fibrils, and propose models for these interactions. We gained structural and mechanistic insights into the interaction of PFD with two different states of amyloidogenic IAPP (monomeric and fibrillar) by combining solution NMR, EM, AFM, and kinetic measurements.

NMR interaction studies report on binding between monomeric IAPP and PFD by chemical shift perturbations and peak broadening on IAPP. Analysis of these effects allowed us to identify two segments on IAPP, one towards the N-terminus (10–19) and one in the middle segment (24–28) to be important for the interaction (Fig. 3). Extensive deuteration and methyl labeling enabled us to study the interaction from the perspective of the 86.4 kDa PhPFD. PRE-mapping with a paramagnetic-labeled IAPP construct on PFD allowed to determine the sites involved in IAPP binding (Fig. 2). As it has been previously observed for chaperone-client complexes, such as the one between Hsp90 and tau protein, the complex is characterized by a broad interaction surface[53]. IAPP induced PRE-broadening effects are observable inside the whole PFD cavity. Especially strong broadening is observed towards the middle of the coiled-coil helices of the α-subunit, suggesting that this is an important region for binding. The apparent $K_D$ was determined to be of intermediate strength (between 11 and 61 μM, according to BLI

and NMR investigation). The extensive binding interface and multiple binding regions on IAPP suggests avidity-based binding to multiple lower affinity contacts which can be sampled by IAPP. The spectra are characteristic of fast exchange between bound and free form with an apparent lifetime of the bound state on the order of 100 to 200 μs. As typical for chaperone-client complexes[52], the interaction between monomeric IAPP and prefoldin cannot be described as one static complex, but as a structural ensemble where IAPP dynamically binds and unbinds the available interaction sites inside the prefoldin cavity. Possible representative models of this complex were obtained by docking of IAPP on PFD driven by NMR derived structural restraints (one of the computed models is shown in Fig. 5a, b). De novo ThT-fluorescence assays show that this interaction between IAPP monomers and prefoldin does only lead to a weak effect on the lag-time. Probably, neither restructuring effects on IAPP, nor sequestration of enough IAPP monomers in PFD cavity occurs, to strongly impact the primary nucleation step, and therefore the PFD-IAPP monomer interaction would play a minor role in the inhibition mechanism of IAPP fibril formation. However, the NMR studies between monomeric IAPP and PFD bring important information about residues of IAPP and PFD susceptible to be involved in the interaction with different species during the IAPP fibrillation pathway.

The ThT-fluorescence assays (Fig. 1) show inhibition of IAPP fibrillation at substoichiometric PFD ratios, which suggests interaction between PFD and high molecular weight IAPP species (e.g., oligomers or fibrils). Further investigation of fibrillation pathways by seeded ThT assays showed that both elongation and secondary nucleation pathways were inhibited by the presence of PFD (both human and archaeal). The subsequently performed EM analysis allowed to image PFD bound to ends and surface of IAPP fibrils (Fig. 4), which reveals that the observed inhibition of the elongation and secondary nucleation pathways is achieved by these interactions (Fig. 5c). Only a low substoichiometric PFD:IAPP ratio is required to obtain a major effect on the fibrillation kinetics, when inhibition is mediated by binding to fibrillar species[57]. The structures of fibrillar IAPP were elucidated recently by cryo-EM[11–15]. One major polymorph observed contains two protofilaments with three β-strands each and has a pitch of 48 nm[13–15]. The fibril core is composed of residues 13–37 with residues 1–12 protruding from the fibril core which are therefore not resolved in the cryo-EM map due to their dynamic behavior. This polymorph (PDB: 6Y1A) was used to model the PFD-IAPP-fibril interaction as observed by EM, utilizing interacting IAPP and PFD residues identified by NMR to dock PFD and the fibril. This allowed us to get a sub-molecular insight into the interaction observed at low resolution by negative stain EM. From the fibril structure, it is clear that only the N-terminal binding epitope determined on IAPP is available for PFD binding when a fibril is formed. The PFD-IAPP fibril interaction through the N-terminal epitope was confirmed by the model obtained by NMR-guided docking[55,56] (Fig. 5d, e). Yet, ca. up to six N-termini of IAPP in its fibrillar state can fit into the PFD cavity when PFD is binding to the fibril surface or end, possibly leading to a higher affinity as

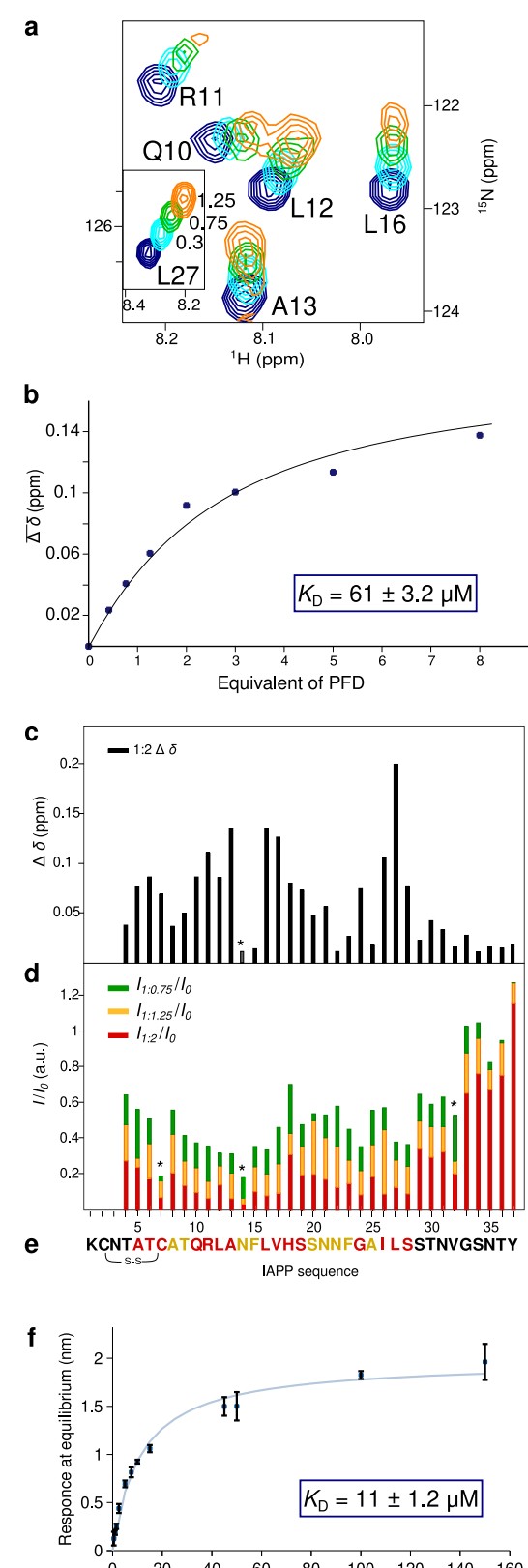

**Fig. 3 Interaction of monomeric IAPP with Prefoldin. a** Extracts of 2D $^{15}$N-SOFAST-HMQC spectra of IAPP titration by addition of PhPFD. [PhPFD]:[IAPP] molar ratios ranging of 0, 0.3, 0.75, and 1.25 are shown. **b** $K_D$ was calculated by line shape fitting of signals of A13, V17, H18, S19, F23, A25 and L27. Here, fit shown against average of all peaks chemical shift perturbations where peak position could be determined. **c** Summary of detected chemical shift perturbations (CSPs) $\Delta\delta$ (calculated as $\Delta\delta = \sqrt{0.14*\Delta N^2 + \Delta H^2}$)[86-88] induced by addition of a two-fold excess of PFD on uniformly $^{15}$N-labeled sample of IAPP. **d** Peak broadening of IAPP signals upon addition of PhPFD, calculated as the intensity ratio of NH signals in IAPP/PhPFD complex ($I$) and in absence of PhPFD ($I_0$). Asterisk denotes values obtained at 10 °C due to peak overlap at 30 °C. **e** IAPP sequence with color coding of detected CSPs at ratio 1:2 (residue with above average $\Delta\delta$, that is $\geq$0.12 ppm in red) and line broadening at ratio 1:2 (residue with $I/I_0 \leq 0.26$ in yellow). **f** Steady state analysis of biolayer interferometry (BLI) experiment on PhPFD binding to monomeric IAPP: measured values (dark blue) and fit (light blue). Data is shown as mean ± SD out of $n = 3$ independent replicates.

important for interaction are exposed at the fibril end. The preference of PFD to bind to fibril ends correlated with the strong inhibition of fibril elongation observed in IAPP aggregation assays (Fig. 1).

A striking effect of PFD on the IAPP fibrillation curves monitored by ThT FI was a decrease of the plateau height with addition of PFD (Fig. 1). This could be due to a change in the fibrillar structure of IAPP leading to different binding of ThT molecules and subsequently decreased ThT FI, shielding of the surface due to PFD decoration, or due to the formation of less IAPP fibrils. A control experiment revealed that pure shielding of the ThT-IAPP fibril surface interaction by PFD is not sufficient to explain the amplitude of the observed effect (see Methods section). However, AFM samples taken directly after termination of fibrillation assays show indeed decreased amounts of fibrils, but simultaneously a morphological change in the observed aggregates is noticed. The fibrils which have formed in presence of PFD are shorter and cluster together, forming larger aggregates. The height profile and observed pitch, which can be measured on fibrils protruding from these aggregates in the AFM images, however are similar to the major polymorph formed in the absence of PFD. It seems that both decreased formation of aggregates and the formation of bigger clusters of aggregates, possibly also decorated by PFD molecules, lead to the observed effect in the ThT assays. The presence of PFD changes the amount and the length of fibrils and leads to their clustering. In this context, it is worth noting that in the literature reduction of toxicity was observed upon decrease of the total exposed hydrophobicity by increase of aggregate size and surface shielding[58,59].

PFD increased the viability of IAPP-treated cultured rat pancreatic beta cells in an MTT assay (Fig. 1c). For this cytoprotective effect to prevail in vivo, PFD and IAPP would need to colocalize in a common cellular compartment. While PFD is localized in the cytosol, IAPP is processed from its prohormone in the Golgi and in the insulin secretory granule, from where it is secreted[60]. Thus, freshly processed and secreted IAPP is unlikely to be a PFD client. However, IAPP oligomers might escape from the secretory pathway, leading to cytosolic IAPP aggregates[60,61]. Moreover, extracellular IAPP oligomers have been shown to be taken up by beta cells, resulting in accumulation of cytoplasmic IAPP and altered cellular proteostasis[62,63]. Such cytosolic IAPP species might well constitute PFD clients.

In conclusion, in this study we demonstrated that both human and archaeal PFD are able to inhibit fibril formation of the

compared to monomer binding. On the fibril end, the second binding site in the middle IAPP segment is additionally available to stabilize the interaction. According to EM, the PFD density at the fibril end was higher than on the lateral fibril surface. This suggests that the fibril end has a particularly high affinity for PFD, which can be explained by the fact that both IAPP segments

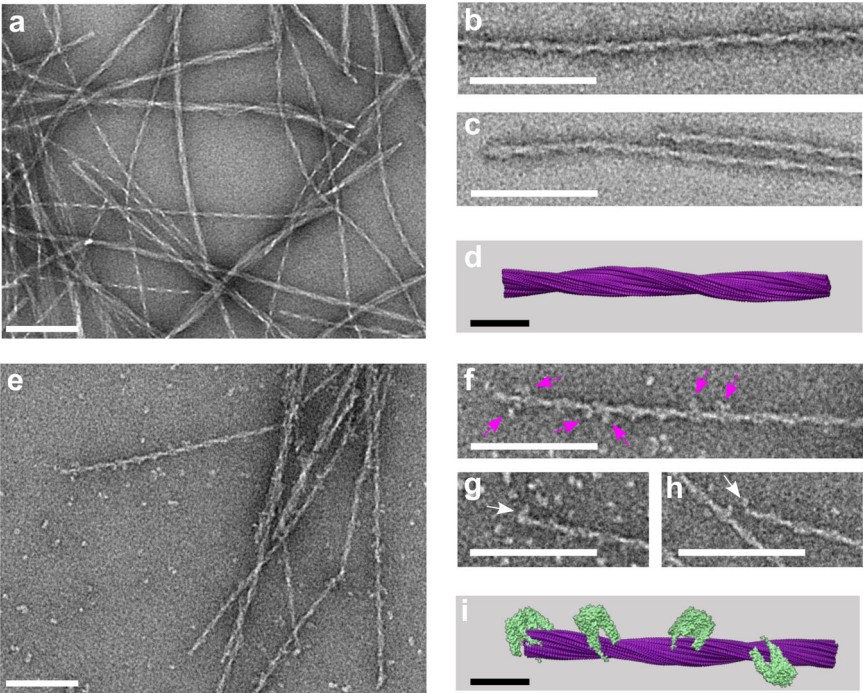

**Fig. 4 EM images of PhPFD bound to IAPP fibrils. a** Electron microscopy images of IAPP fibrils, stained with sodium silicotungstate (SST) at 30,000x magnification. **b**, **c** Close-ups of main fibril species. **d** Molecular model of polymorph 1 of the IAPP fibril (PDB: 6Y1A)[13]. **e** Preformed IAPP fibrils (67 µM, monomer equivalent concentration) were incubated with PhPFD in a ratio of 100:1 for 30 min and imaged under the same conditions as IAPP fibrils (**a**). Decoration of the fibril surface with PhPFD is observed, remaining PhPFD particles are observed in the background. **f–h** Close-ups of the fibrils show the decoration of the fibril surface (pink arrows) and end (white arrows). **i** Molecular model corresponding to the observed fibril decoration. Horizontal white and black bars correspond to a length of 100 nm and 10 nm, respectively. Simulated TEM images of (**d**, **i**) are shown in Supplementary Fig. 10.

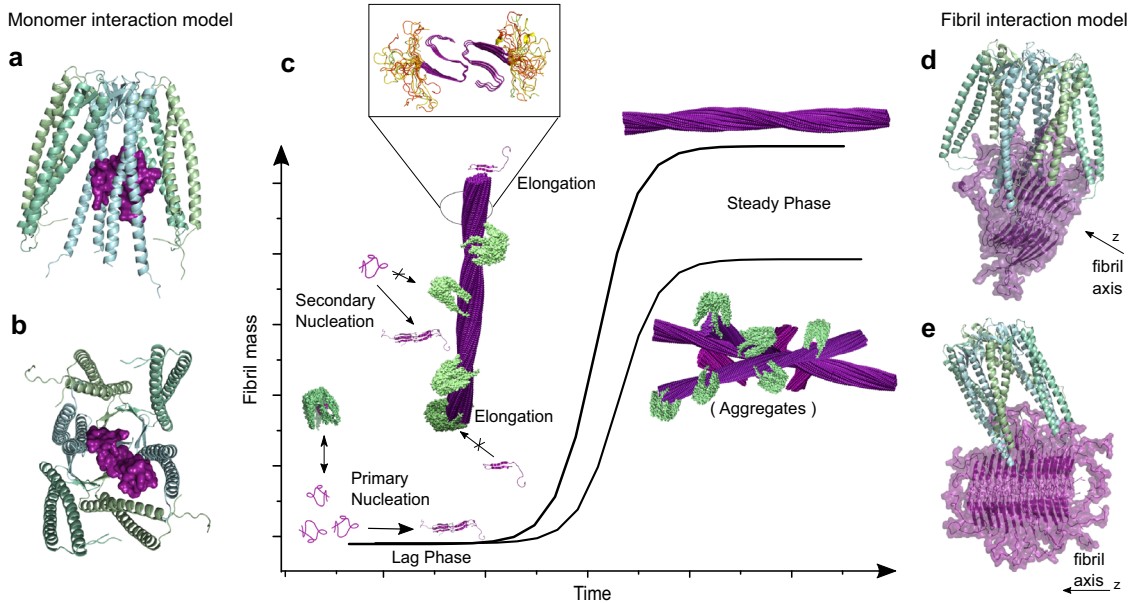

**Fig. 5 Structural Model of inhibition of IAPP fibril formation by PFD.** Schematic model describing the inhibition mechanism of PFD on IAPP fibril formation. PFD interacts with monomeric IAPP, but this transient interaction does not lead to a significant decrease of the lag-phase. **a**, **b** present docking models of the complex between monomeric IAPP (PDB: 2L86) and PhPFD (PDB: 2ZDI) based on NMR derived interaction information (Figs. 2 and 3). **c** Inhibition of secondary nucleation and elongation results from coverage of fibril surface and ends by PFD (Fig. 4). The presence of PFD leads to a decreased steady phase fibril mass (Fig. 1a), which results from the formation of less aggregates with an altered morphology (Fig. 1b). The inset zoom represents a model of IAPP fibril structure with unfolded residues 1–12 in yellow and the structured fibril core is represented in purple (from residues 13–37). **d**, **e** present docking models of PhPFD (PDB: 2ZDI) on IAPP fibril (PDB: 6Y1A) surface and extremities, respectively, integrating structural information obtained by NMR (Figs. 2 and 3) and EM (Fig. 4); the black arrows indicate the fibril axis with the tips pointing towards the fibril ends.

amyloidogenic protein IAPP. Solution NMR enabled us to identify two binding regions on IAPP (from residues 10 to 19 and 24 to 28), and to map the corresponding binding sites within the PhPFD cavity. We established, using IAPP fibrillation assays and electron microscopy, that the substoichiometric inhibition of IAPP fibrillation by PFD is mainly due to binding to fibril surface and ends, thereby inhibiting both secondary nucleation and elongation. Binding to the fibril is mediated by the N-terminal regions of IAPP of which up to six can be enclosed in the PFD cavity, interacting with PFDs coiled-coil helices. Binding to fibril ends is possibly supported by the second binding region, located in the middle segment of IAPP. The presence of PFD leads to the formation of lower amounts of aggregates, composed of shorter fibrils and clustered into formations of bigger size, which could be a potential detoxifying mechanism.

## Methods

### Protein preparation

*IAPP.* Human IAPP (H-KCNTATCATQ RLANFLVHSS NNFGAILSST NVGSNTY-NH$_2$; molecular mass 3903.4 Da; purity 93.2%), biotinylated IAPP (biotinyl-[β-Ala]-[β-Ala]-KCNTATCATQ RLANFLVHSS NNFGAILSST NVGSNTY-NH$_2$; molecular mass 4271.9 Da; purity 99.2%), and DOTA (2,2′,2″,2‴-(1,4,7,10-Tetraazacyclododecane-1,4,7,10-tetrayl)tetraacetic acid) labeled IAPP (DOTA-[β-Ala]-[β-Ala]-KCNTATCATQ RLANFLVHSS NNFGAILSST NVGSNTY-NH$_2$; molecular mass 4432.4 Da; purity 95.9%), all with an amidated C terminus and a disulfide bond between Cys2 and Cys7 were custom synthesized (Pepscan, Lelystad). Identity and purity were confirmed by RP-HPLC and mass spectrometry.

For kinetic experiments, to ensure monomeric starting material, the IAPP peptide powder was dissolved at 2 mg/ml in 1,1,1,3,3,3-hexafluoro-2-propanol (HFIP), incubated at room temperature for 1 h and lyophilized. Then 1 mg peptide powder was dissolved in 0.5 ml aqueous 6 M guanidine hydrochloride solution, and size-exclusion chromatography was performed on a Superdex 75 Increase 10/300 column (GE Healthcare) equilibrated with 10 mM 2-(N-morpholino)ethanesulfonic acid (MES)/NaOH buffer, pH of 6.0 using an NGC liquid chromatography system (Bio-Rad). The monomer peak fraction was collected, aliquoted, flash frozen in liquid nitrogen, stored at −80 °C, and thawed on ice straight before further use.

DOTA-functionalized IAPP was loaded with either paramagnetic Gd$^{3+}$ or diamagnetic Lu$^{3+}$ cations, respectively. Lyophilized DOTA-[ß-Ala]-[ß-Ala]-IAPP-NH$_2$ (400 μg, 90.2 nmol) was dissolved in 500 μl 20 mM sodium acetate/acetic acid buffer pH 4.53 and 18 μl 500 mM aqueous GaCl$_3$ solution was added (9 μmol, i.e., a 100-fold molar excess). The solution was incubated at 25 °C and 300 rpm shaking. Time dependent Gd$^{3+}$ loading was monitored by analyzing the reaction mixture from withdrawn 10 μl aliquots at different time points by RP-HPLC on a 9.4 mm × 250 mm Zorbax 300SB-C8 column connected to a 1260 HPLC system (both Agilent) operated at a column temperature of 80 °C, a flow rate of 4 ml/ml and UV absorbance monitoring at 214 nm. A linear gradient from aqueous 8% (v/v) acetonitrile (ACN), 0.1% (v/v) trifluoroacetic acid (TFA) to 60% (v/v) ACN, 0.1% (v/v) TFA within 40 min was applied. Gd$^{3+}$ loading was completed after ~2 h and the remaining reaction mixture was subsequently purified by two to three consecutive preparative RP-HPLC runs (each 30–50 nmol Gd$^{3+}$-DOTA-IAPP) under elution conditions described above.

Time dependent RP-HPLC analytics are shown in Supplementary Fig. 4a. Loading of DOTA-IAPP with Lu$^{3+}$ was performed in an analogous way (Supplementary Fig. 4b).

U-[$^{15}$N]- or U-[$^{15}$N, $^{13}$C] isotopically labeled human IAPP was recombinantly expressed in *Escherichia coli* in M9 medium supplemented with either 2 g/l $^{15}$N-NH$_4$Cl and 3.2 g/l unlabeled glucose or with 2 g/l $^{15}$N-NH$_4$Cl and U-[$^{13}$C] glucose and purified according to established protocols[64], resulting in production of IAPP in its non-amidated form. Purified recombinant IAPP from preparative RP-HPLC as final purification step was batch lyophilized, dissolved in HFIP, aliquoted and lyophilized again. Identity and purity (93%) were confirmed by analytical RP-HPLC under isocratic conditions with 28.5% (v/v) ACN, 0.1% (v/v) TFA as mobile phase on a 4.6 mm × 250 mm Zorbax 300SB-C8 column (Agilent) at 80 °C column temperature and 1 ml/min flow rate (Supplementary Fig. 4c). For quality control a 2D $^{15}$N-TROSY spectrum of U-[$^{15}$N]-IAPP was recorded at 10 °C on a NMR spectrometer with a $^{1}$H frequency of 850 MHz (Supplementary Fig. 4d).

*Human Prefoldin (hPFD – 97.0 kDa).* E. coli Rosetta2 (DE3) pLysS cells (Novagen) were transformed with pET-21a plasmids encoding for PFD1, PFD3, PFD5 and PFD6[65] or pET-41 (Genecust) encoding for PFD2 and PFD4 subunits of hPFD. The N-terminal 37 residues of PFD3 and four residues of PFD4 were deleted[65], PFD2 and PFD4 contained N-terminal hexahistidine tags followed by a thrombin cleavage site (Supplementary Fig. 7a). Bacteria were either transformed with the plasmids for PFD1, PFD3, PFD5 and PFD6 individually or co-transformed with

the plasmids for PFD2 and PFD3. The cells were grown at 30 °C in LB-medium with the required antibiotics (Chloramphenicol, Ampicillin, or Kanamycin). At OD$_{600nm}$ of 0.6 protein production was induced by IPTG (1 mM) and the expression was performed at 20 °C for 8 h (PFD1, PFD5), 30 °C for 4 h (PFD2, PFD3, PFD4) or 30 °C for 2 h (PFD6). All purification steps were carried out at 4 °C. Cells expressing the different subunits were mixed (100 ml for PFD1, PFD5 or PFD2/PFD3, 150 ml for PFD6 and 200 ml for PFD4). Cells were sonicated in 50 ml of 50 mM sodium phosphate buffer (pH 7), containing 300 mM NaCl, 5 mM β-mercaptoethanol and complemented with 0.025 mg/ml RNAse (Euromedex), 0.025 mg/ml DNAse (Sigma Aldrich) and 1 anti-protease tablet (cOmplete™). After removal of cell debris by centrifugation, affinity purication was performed on a HisTALON™ column, equilibrated with 50 mM sodium phosphate buffer (pH = 7), containing 300 mM NaCl and 5 mM beta-mercaptoethanol[65]. The protein was eluted with 150 mM imidazole. Fractions containing hPFD were combined and his-tag removal was performed over night (16 h), using 8 units of thrombin (GE Healthcare, 27-0846-01) per mg of protein under shaking at 4 C. The sample was loaded on a MonoSTM 5/50 GL column equilibrated with 20 mM MES (pH = 6), 1 mM DTT. The protein was eluted at 20% of 20 mM MES (pH = 6), 1 mMDT, 1 M NaCl. Size exclusion chromatography was performed on a S00 30/100 column equilibrated with 20 mM Tris (pH = 7.5), 150 mM NaCl, 1 mM DTT[65]. SEC-MALS and SDS-PAGE followed by Coomassie staining was used to monitor the purification process (Supplementary Fig. 6b, c).

*Prefoldin from* Pyrococcus horikoshii (PhPFD – 86.4 kDa). E. coli BL21 (DE3) cells transformed with pET23c plasmids encoding either for the α- or β-subunit of PFD from *Pyrococcus horikoshii* (point mutation on α-subunit S98G) were used for protein expression (Supplementary Fig. 8a). To obtain unlabeled subunits, cells were grown at 37 °C in LB-medium, at OD$_{600nm}$ of 0.8 protein production was induced by IPTG (1 mM) and the expression was performed at 37 °C for 3 h. For production of U-[$^{2}$H, $^{13}$C, $^{15}$N]-PhPFD samples, cells were progressively adapted to M9/$^{2}$H$_2$O media in three stages over 24 h. In the final culture the bacteria were grown at 37 °C in M9 media prepared with 99.85% $^{2}$H$_2$O (Eurisotop), 2 g/l of U-[$^{2}$H,$^{13}$C] D-glucose and 1 g/l $^{15}$NH$_4$Cl (Cambridge Isotope Laboratories). For production of U-[$^{2}$H, $^{12}$C, $^{15}$N], A-[$^{13}$CH$_3$]$^{β}$, I-[$^{13}$CH$_3$]$^{δ1}$, L-[$^{13}$CH$_3$]$^{δ2}$, V-[$^{13}$CH$_3$]$^{γ2}$, T-[$^{13}$CH$_3$]$^{γ2}$ labeled subunits, 2 g/l of U-[$^{2}$H] D-glucose (Sigma Aldrich) was used as carbon source. When the OD$_{600nm}$ reached 0.7, a solution containing 240 mg/l of 2-[$^{13}$C]methyl-4-[$^{2}$H$_3$]-acetolactate (NMR-Bio) was added in M9/$^{2}$H$_2$O media for the stereoselective labeling of pro-S Leu$^{δ2}$ and Val$^{γ2}$ methyl groups[66]. 40 min later 3-[$^{13}$C]-2-[$^{2}$H]-L-Alanine, (S)-2-hydroxy-2-(2′-[$^{13}$C],1′-[$^{2}$H$_2$])ethyl-3-oxo-4-[$^{2}$H$_3$]-butanoic acid (NMR-Bio) and 2,3-($^{2}$H) 4-($^{13}$C)-L-Threonine (NMR-Bio) were added to a final concentration of 250 mg/l, 100 mg/l and 50 mg/l respectively[67,68] for the simultaneous labeling of Ile$^{δ1}$, Ala$^{β}$, Thr$^{γ2}$ methyl groups. Protein production was induced by IPTG (1 mM) and protein expression was performed at 37 °C for 3 h. With such protocol, the level of deuteration of protein is higher than 97%, while selectively labeled methyl probes are estimated to be protonated at more than 95%.

The purification protocol of archaeal prefoldin subunits was adapted from Okochi and coll.[69]. SDS-PAGE followed by Coomassie staining, SEC-MALS and cryo-EM was used to control purity and homogeneity of produced samples (Supplementary Fig. 7b–d). Mass spectrometry was performed on the α-subunits (Supplementary Fig. 8f). For interaction studies, the buffers of PhPFD-α$_2$β$_4$ samples were exchanged for 25 mM MES/NaOH buffer (pH 6.5), 25 mM of MgCl$_2$ using Amicon Centrifugal Filter Units (Merck).

**Thioflavin T (ThT)-fluorescence assays**. IAPP aggregation kinetics were monitored by ThT fluorescence intensity (FI). All the experimental solutions contained 5 μM monomeric IAPP, 10 μM ThT, and 6 mM NaN$_3$ in 50 mM MES/NaOH (pH 6.5), 25 mM MgCl$_2$. Each assay contained a row of IAPP:PFD ratios: a sample without PFD and 10 samples with different PFD concentrations, corresponding to logarithmically equidistant molar ratios PFD:IAPP between 1:5000 and 1:1. De novo assays had no additional components. For seeded assays, pre-formed mature IAPP fibrils were added (8–9% of overall IAPP concentration in monomer equivalent): either non-sonicated in the secondary nucleation assay in order to provide the fibril surface capable of catalyzing nucleation[33], or sonicated (Sonopuls MS 72 microtip sonotrode, Bandelin; 10% amplitude, 4 cycles of 1 s pulse and 5 s break) in the elongation assay, thus providing a higher number of fibril ends to recruit and incorporate monomers resulting in the linear growth of fibril mass[44]. For all assays, the samples were prepared as triplicates in protein LoBind tubes (Eppendorf) on ice and then transferred into a 96-well half-area polystyrene non-binding surface (NBS) microplate, black with flat transparent bottom (3881, Corning). ThT FI values were recorded in a BMG FLUOstar Omega microplate reader controlled by Reader Control software from BMG (BMG LABTECH, Ortenberg, Germany) at 37 °C, using excitation wavelengths of 448-10 nm and emission of 482-10 nm, each data point corresponds to an orbitally averaged value over 3 mm with 20 flashes per well. In the de novo assays, 20 s orbital shaking at 300 rpm before each cycle was applied as additional agitation.

Kinetic analysis was performed and various parameters were extracted using numerical approximations of first and second derivatives with respect to time of the ThT FI curve (Fig. 1a, Supplementary Fig. 1a), as described below. For the

analysis of the elongation assays, only the first linear parts of the curves were considered, since under the chosen conditions other aggregation mechanisms besides elongation start contributing to IAPP fibrillation from relatively early time-points.

Since in all kinetic assays the resulting final ThT FI plateau values were significantly lowered upon increasing PFD concentration, a control experiment was performed: after the completion of an aggregation assay, PFD concentrations were equalized in all the wells and then ThT FIs were re-measured. The difference in the plateau heights was preserved as before this manipulation. Hence, the change in the final plateau values cannot be simply attributed to PFD preventing ThT alignment on the fibrillar surface and its following fluorescence enhancement, but it rather reflects the amount of available fibrillar IAPP in solution. For this reason, the kinetic analysis shown in Fig. 1a was performed on non-normalized data. However, the comparison of the changes in kinetic parameters extracted from normalized and non-normalized data showed highly similar tendencies (Supplementary Fig. 1a), indicating the reliability of the conclusions on PFD inhibitory effects on IAPP aggregation.

Experimental data shown in Fig. 1 for de novo and secondary nucleation assays were lag-time-corrected before averaging over the triplicates: lag time for each replicate was set to the mean lag time of the according triplicate by a slight shift of unmodified curves along the time axes. Lag time, defined here as the time before the first order time derivative reaches 5% of its maximum value, varied in triplicates on average about 6%. An exemplary comparison of modified and non-modified data is shown in Supplementary Fig. 1b.

Extracting the kinetic parameters was done automated using a custom script, processing the data uniformly. First, data was smoothened via moving averaging over 11 data points. Then, numerical approximations of first and second order time derivatives were calculated as the symmetric difference quotient of the ThT FI data and first order derivative accordingly, using a derivative depth window of 4 points. Growth phase duration was defined as time between growth start and end (time points where first derivative has a value of 5% of its maximum); maximal growth rate – as maximal value of first order derivative; average growth rate – as the difference between ThT FI values at end and start of growth, divided by the growth phase duration; initial growth rate was calculated as mean value of first 20 values of the first order derivative; final plateau height – as a mean of 50 points following after the growth end; acceleration maximum – as the maximal value of second order time derivative. The full text of the used script is linked to this publication.

**AFM imaging**. For the AFM imaging, after recording de novo aggregation assays, the samples were taken out of the plate (96-well Half Area Black/Clear Flat Bottom Polystyrene NBS Microplate; 3881, Corning). The solution was extensively mixed in order to include the aggregates potentially sticking to the walls of the well. 2 µl of each sample were put onto a freshly cleaved muscovite mica surface and dried during the incubation over 10 min under the clean bench. Subsequently, the samples were washed 3 times with 100 µl Milli-Q $H_2O$ and dried with a steam of $N_2$ gas. Imaging was performed in intermittent contact mode (AC mode) in a JPK NanoWizard 3 atomic force microscope equipped with NanoWizard Control Software v.5 version 5.0.84 by JPK (JPK, Berlin, Germany) using a silicon cantilever with silicon tip (OMCL-AC160TS-R3, Olympus) with a typical tip radius of $9 \pm 2$ nm, a force constant of 26 N/m and a resonance frequency around 300 kHz. For each sample a net area of 2500-18400 µm$^2$ was scanned for the overview, while part of it was additionally imaged with a higher resolution.

AFM imaging in liquid was done as well on the samples after a de novo aggregation assay. For that, 10 µl of each sample for the microplate were put onto a freshly cleaved muscovite mica surface positioned inside a reservoir for liquid on a glass slide, and incubated under a humid atmosphere for 20 min. Subsequently, the reservoir was filled with 1 ml of 50 mM MES/NaOH (pH 6.5), 25 mM MgCl$_2$. Imaging was performed in intermittent contact mode (AC mode) in a JPK NanoWizard 3 atomic force microscope (JPK, Berlin) using a silicon nitride cantilever with silicon tip (SNL-10-A, Bruker) with a nominal tip radius of 2 nm, a spring constant of 0.35 N/m and resonance frequency around 65 kHz.

The images were processed using JPK DP Data Processing Software (version spm-5.0.84). For the presented height profiles, a polynomial fit was subtracted from each scan line, first independently and then using limited data range.

**Cell viability assay (MTT test)**. In order to examine the cytotoxic effects of IAPP with or without (different) concentrations of PhPFD, an MTT (3-(4,5-Dimethyl-thiazol-2-yl)-2,5-diphenyl-tetrazolium bromide) based cell viability assays was accomplished with RIN-m5F cells (rat pancreatic beta cells).

RIN-m5F cells (ATCC, CRL-11605, Manassas, VA, USA) were cultured in RPMI 1640 medium (Thermo Fisher ATCC modification, Germany) supplemented with 10% fetal calf serum, 1% antibiotics (Penicillin/Streptomycin) (all Sigma-Aldrich, Germany) on tissue culture flasks (Avantor, Germany). According to their confluence, cells were passaged every three to five days and were allowed to grow in a humidified incubator with 5% CO$_2$ at 37 °C.

The MTT test was conducted according to the manufacturer's protocol (Cell Proliferation Kit I; Sigma Aldrich, Germany). In brief, at a density of $2.7 \times 10^6$ in 100 µl complete medium, RIN-m5F cells were seeded in clear 96-well flat bottom microwell plates (Life Technologies Inc., USA) and cells were allowed to adhere and grow for 24 h. After 24 h, compounds were prepared in the following way and

added to the cells (four to five technical replicates): monomeric IAPP at a concentration of 51 µM in 50 mM MES/NaOH (pH 6.5), 25 mM MgCl$_2$ was placed in protein lo-bind Eppendorf tubes without PFD or with PhPFD at PFD:IAPP molar ratios of 0.1, 0.3, and 1, and quiescently incubated for 3 days at room temperature. Before addition to cells, all samples were mildly sonicated for 5 min in an ultrasonic bath (Bandelin SONOREX RK 100 H) and extensively mixed via pipetting. Cells were exposed to 1 µM IAPP with or without PhPFD. Triton X100 served as a negative control. After an additional incubation time of 24 h, the MTT labeling regent (10 µl) was added to each well and incubated for 4 h. Next, the solubilization buffer (100 µl) was added and incubated overnight. The next morning, the microplate was evaluated using a plate reader (CLARIOstar, BMG LABTECH, Ortenberg, Germany) at 550 to 600 nm with a reference wavelength of >650 nm. Results are represented as the percentage of MTT reduction, assuming that the absorbance of control cells (medium) was 100%.

**Biolayer interferometry (BLI)**. The BLI measurements were conducted using BLItz-System equipped with BLItz Pro 1.2.1.5 software by ForteBio (FortéBio, Sartorius), at room temperature. Super Streptavidin (SSA) biosensors (Sartorius) were hydrated in 50 mM MES/NaOH (pH 6.5), 25 mM MgCl$_2$ buffer, loaded with biotinyl-[β-Ala]-[β-Ala]-IAPP-NH$_2$, quenched with biotin to eliminate the non-specific binding of PhPFD to biosensors, and equilibrated in buffer. The sensors were plunged in solutions containing different concentrations of PhPFD from 0.5 µM to 150 µM and the association curves were recorded over 180 s, followed by the dissociation in the according buffer for 240 s. The resulting binding kinetics were corrected using the blank reference, then a steady state analysis was performed using $R_{eq} = \frac{[A]R_{max}}{([A]+K_D)}$ fitting equation, where $R_{eq}$ and $R_{max}$ stay for equilibrium and maximum responses, and $[A]$ is analyte (PhPFD) concentration.

**NMR spectroscopy**

*IAPP-PFD titration*. The NMR titration samples were prepared by adding unlabeled PhPFD into solutions of U-[$^{15}$N]-labeled IAPP. Lyophilized, purified monomeric IAPP was suspended in an ice-cold buffer (25 mM MES/NaOH (pH 6.5), 25 mM MgCl$_2$) at a concentration of 64 µM and aliquoted. The aliquots were flash-frozen in liquid nitrogen and stored at −80 °C. For every titration point the aliquots were thawed on ice; different amounts of PFD, buffer and 4 µl of $^2$H$_2$O were added, so that the final concentration of IAPP was set to 29 µM (total volume 44 µl). The ratios: 1:0 (apo), 1:0.4, 1:0.75, 1:1.25, 1:2, 1:3, 1:5 and 1:8 were tested. Each sample was transferred into a 1.7 mm NMR tube and 2D $^{15}$N-SOFAST-HMQC spectra[70] were acquired at 30 °C on a Bruker Avance III HD spectrometer equipped with a 1.7 mm cryogenic probe and operating at a $^1$H frequency of 850 MHz. For 2D acquisition time with an acquisition time longer than 5 h, the NMR data acquisitions were split to control that the IAPP NMR signals did not decrease over time, ensuring that the IAPP state did not change over the course of the NMR experiment. Data were acquired using TopSpin (Bruker) and NMRlib[71] processed using nmrPipe/nmrDraw[72], CcpNmr and the $K_D$ and $k_{off}$ were extracted using TITAN global lineshape fitting[54] using NMR signals of residues A13, V17, H18, S19, F23, A25 and L27, assuming a 1:1 stoichiometry, therefore using the two-state binding model. IAPP was assigned using 3D BEST-TROSY HNCO, HNCA, HN(CO)CACB, HN(CA)CB and HN(CA)CO[71] experiments (BMRB accession code 51259).

*PhPFD assignment*. Backbone atoms and methyl groups resonances of both α- and β-PhPFD subunits were previously assigned[45], using a combination of (i) 3D BEST-TROSY HNCO, HNCA, HN(CA)CB, HN(CO)CA, HN(COCA)CB and HN(CA)CO experiments[71]; (ii) structure-based analysis of inter-methyl NOEs[73]; (iii) mutagenesis[74] and (iv) 3D HCC-relay experiments[75,76] connecting backbone to methyl moieties. List of assigned chemical shifts is available online (BMRB accession code 50845) and annotated 2D $^{13}$CH$_3$- and $^{15}$N-TROSY spectra (70 °C in 50 mM Tris (pH 8.5), 100 mM NaCl) are presented on Supplementary Fig. 3.

*PRE-experiments*. $^{15}$N- or $^{13}$CH$_3$-labeled U-[$^2$H]-PhPFD sample was combined at a ratio of 1:1 (or 1:2 for methyl labeled samples) with DOTA-modified IAPP loaded with Lu$^{3+}$ (diamagnetic) or Gd$^{3+}$ (paramagnetic) and transferred in 4 mm Shigemi tubes. For [$^{15}$N, $^2$H]-PhPFD the resulting concentrations were 100 µM PFD and 100 µM IAPP, for $^{13}$CH$_3$-labeled U-[$^2$H]-PhPFD ~50 µM PFD and 100 µM IAPP. For each sample 2D $^{15}$N-BEST-TROSY[77] or 2D SOFAST-METHYL-TROSY spectra[74] were recorded at 30 °C for ~0.5 day/sample on a Bruker Avance III HD spectrometer equipped with a cryogenic probe and operating at a $^1$H frequency of 950 MHz. For long PRE experiments, the NMR data acquisitions were split and 1D control NMR experiments were inserted before and after each 2D experiment. These control experiments were enabling to monitor IAPP NMR signals over time, ensuring that the IAPP state did not change over the course of the NMR experiment. As a control experiment to exclude false positive interaction of PFD with the DOTA-cycle, we verified that addition of Gd$^{3+}$-loaded DOTA to methyl-labeled PFD does not lead to specific residue broadening (Supplementary Fig. 9).

**Electron microscopy (EM)**. IAPP fibrils were formed by incubation of 67.4 µM monomeric IAPP at room temperature for a minimum of one week in 10 mM

MES/NaOH (pH 6.0), 6 mM NaN$_3$. Concentrated purified PFD in 25 mM MES/NaOH (pH 6.5) 25 mM MgCl$_2$ was added in a ratio of 1:100 (PFD:IAPP-monomer equivalent concentration) and incubated for 30 min. Samples were adsorbed to the clean side of a carbon film on mica, stained with sodium silicotungstate (SST) Na$_4$O$_{40}$SiW$_{12}$ at 1% (w/v) in distilled water (pH 7–7.5) and transferred to a 400-mesh copper grid[78]. The images were taken under low dose conditions (<30 e$^-$/Å$^2$) with defocus values between $-1.2$ and $-2.5$ μm on a FEI Tecnai 12 LaB6 electron microscope at 120 kV accelerating voltage, 30,000x nominal magnification using CCD Camera Gatan Orius 1000. Images were acquired using Digital Micrographs 1.85.1535 and analyzed with Gwyddion software[79].

For cryo-EM imaging of PhPFD, 3.5 μl of PFD solution in 50 mM Tris, 100 mM NaCl pH 8.5 buffer at 100 μM was blotted on a Quantifoil R2/2300 mesh gold grid coated with carbon and ionized by glow discharge. A FEI Vitrobot (automated vitrification machine) MARK IV was used at 6.5 sec blotting time, 100% humidity at 20 °C.

About 1000 movies were recorded with a Thermo Scientific GLACIOS 200 kV FEG with a Falcon II electron counting direct detection camera with EPU (automatic data collection) with a pixel size of 1.206 Å. About 700,000 particles were extracted using guided particle picking, using the PhPFD crystal structure (PDB: 2ZDI).

2D classification was performed using relion standard procedures[80].

**Modeling of IAPP-PhPFD complexes**. Interaction models were created by HADDOCK rigid model docking[55], using the HADDOCK 2.4 web server[56]. For the model of the complex between monomeric IAPP and PhPFD, the basic docking protocol was used with the interaction residues on IAPP defined as the residues which showed above average chemical shift displacement upon PFD addition for most of the ratios (residues 6, 10-13, 17-19, 24, 26-28). The PhPFD interaction site was defined as residues which showed more than 50% of signal intensity loss upon interaction with paramagnetic-labeled IAPP (α-subunit: 8, 9, 12, 20, 22, 24, 26, 29, 31, 37, 41, 65, 118, 125, 129, 132, 135, 146; β-subunit: 4, 12, 15, 17, 22, 24, 26, 34, 46, 58, 81; β'-subunit: 4, 12, 15, 17, 22, 24, 26, 34, 58, 61, 81, 107; Supplementary Fig. 5). The solution structure of IAPP in SDS micelles (PDB: 2L86) determined by NMR[81] was used as the starting IAPP monomer conformation. Residues on the N-terminus of β-subunits missing in the PhPFD crystal structure (PDB: 2ZDI)[82] were obtained from molecular modeling (MODELLER[83]). For the complex involving PhPFD and IAPP fibril, the cryo-EM structure of the polymorph 1 of IAPP fibrils (PDB: 6Y1A)[13] was used. The 12 missing N-terminal residues were added using molecular modeling (MODELLER[83]), whereby a disulfide bond between residue two and seven was enforced. The same residues in PhPFD-IAPP monomer docking were used as restraints in PhPFD-IAPP fibril docking. For the fibril end interaction model, an IAPP fibril construct of 16 chains was used, and all the according residues on all the chains were given as docking restraints. To direct the docking towards the middle of the fibril, a longer IAPP fibril construct of 26 chains was used, and the interaction restraints were only set on the 18 chains in the middle of the construct. The interacting residues of IAPP on the fibril surface or on the fibril end with PFD residues in the docking models are presented on Supplementary Fig. 11.

**Reporting summary**. Further information on research design is available in the Nature Research Reporting Summary linked to this article.

## Data availability

The NMR assignments data together with corresponding NMR experiments used in this study are available in the Biological Magnetic Resonance Data Bank under accession code 51259 for IAPP[84] and 50845 for PhPFD. The titration and PRE NMR experiments used in this study are available in the Biological Magnetic Resonance Data Bank under accession code bmrbig32. All other experimental data (EM, AFM, BLI, ThT and Cell viability assays) used in this study are available in Zenodo repository under accession code 6319386.

## Code availability

The custom script used to analyze kinetic data is available in Zenodo repository[85] under accession code 6380983.

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

## Acknowledgements

The authors thank Drs. Rida Awad and Leandro Estrozi for help and advice, Drs. K. Miki, M. Fujihashi and J.M. Valpuesta for providing clones of Prefoldin. This work used the high field NMR and EM facilities at the Grenoble Instruct-ERIC Center (ISBG; UAR 3518 CNRS-CEA-UGA-EMBL) within the Grenoble Partnership for Structural Biology

(PSB). IBS platform access was supported by FRISBI (ANR-10-INBS-05-02) and GRAL, a project of the University Grenoble Alpes graduate school (Ecoles Universitaires de Recherche,) CBH-EUR-GS (ANR-17-EURE-0003). The electron microscope facility is supported by the Auvergne-Rhône-Alpes Region, the Fondation pour la Recherche Médicale (FRM), the Fonds FEDER and the GIS-Infrastructures en Biologie Santé et Agronomie (IBiSA). IBS acknowledges integration into the Interdisciplinary Research Institute of Grenoble (IRIG, CEA). This work was supported by the French National Research Agency in the framework of the "*Investissements d'avenir*" program (ANR-15-IDEX-02). RT acknowledges a Ph.D. fellowship from Idex and FRM *(FDT202012010629)*. WH acknowledges support from a European Research Council (ERC) Consolidator grant (no. 726368). LG and DW acknowledge support from the Russian Science Foundation (RSF) (project no. 20-64-46027) for isotopically or PRE-labeled IAPP preparation.

## Author contributions

D.W., J.B., R.T., T.K. and W.H. designed the experiments; R.T., T.K. & L.G. prepared the samples; S.S., L.G. & T.K. performed cell-viability assay, L.G., T.K., R.T. and W.H. performed and analyzed the ThT fluorescence assays, BLI and AFM experiments; E.C.D., J.B., P.G. and R.T. collected and analyzed the NMR experiments; D.F., G.S. and R.T. collected and analyzed the EM data. R.T. modeled the IAPP/PhPFD complexes; J.B., L.G., R.T., T.K. and W.H. wrote the manuscript. All authors discussed the results, corrected the manuscript and approved the final version of the paper.

## Funding

## Competing interests

The authors declare no competing interests.
