## [Peer Review File · Nature Communications]

REVIEWER COMMENTS

Reviewer #1 (Remarks to the Author):

In the manuscript by Törner and coworkers, the researchers subjected amylin fibrils to HSP60, the chaperonin present in the cytosol of archaeal and eukaryotic cells. The researchers determined interactions between the HSP60 and the fibrils using NMR and found, using electron microscopy, that HSPR 60 binds to the fibrils.

This work, nearly entirely repeats the studies published back in 2012 and 2013 by Robb and Lednev: doi: 10.1016/j.bbrc.2012.04.113 and doi.org/10.1021/cb400238a. The researchers didn't cite this work, which suggests that they very poorly reviewed the literature on the topic of their research.

In the light of the previously published work, there is very little if any novelty in the submitted manuscript.

Also, although the structural part of the work is in the good shape, the authors fully overlooked the cell toxicity of the work; they neither evaluate the behavior aspect of their work on animal modes, which is expected for this highly ranked journal.

Reviewer #2 (Remarks to the Author):

The manuscript entitled « structural basis for the inhibition of IAPP fibril formation by the Hsp60 co-chaperonin Prefoldin » examines the effects of HSP60 type II co-chaperonin prefoldin protein (PFD) on the human islet amyloid polypeptide (hIAPP) fibril formation and structural conformation. hIAPP is an amyloidogenic peptide that form fibril in the pancreas of type 2 diabetes patients. In this paper, the authors investigated the hIAPP fibril formation in the presence of PFD using ThT kinetics fluorescence and atomic force microscopy. They examined the hIAPP/PFD interactions using NMR and were able to elucidate the mechanisms of inhibition. This is clearly an interesting paper. The manuscript is well-written. I recommend publication of this paper after the following corrections.

1) My main concern is the images obtained with AFM. The authors stated that the samples are dried. Why did they dry the sample since we know that the drying step induce some changes in the morphologies of the samples? I recommend strongly the authors to perform again the AFM images without the drying step. This is also maybe the reason of the observation of the clustered aggregate of IAPP in the presence of PFD.

2) Some kinetics analysis is not also well described and will be difficult to understand. For example, this is not obvious that the experiment without agitation and with preformed fibrils will give information on the secondary nucleation. Same remark for the experiment with the sonicated, short preformed fibrils on the elongation. I recommend the authors to re-write the part on the kinetics analysis and explain more the analysis and give more references. In addition, the terms "lag time corrected" in figure S1 should be explained.

3) It seems that PFD is found in the cytosol of cells. However, hIAPP is matured in the secretory granules and secreted directly in the extracellular domain. Do you think that hIAPP and PFD could be in the same place in cells?

4) minor corrections:

Do the well plates used in this study are black or transparent? The author should add this information.

The concentration of hIAPP used in the study should be mentioned on the legend of the figures.

Reviewer #3 (Remarks to the Author):

In this paper, R. Törner et al. provide for the first time a molecular basis to understand the mechanism of inhibition of IAPP amyloid fibril formation by prefoldins, which are chaperones belonging to the Hsp60 co-chaperonin family. The results are based on a comprehensive set of complementary techniques, namely ThT fluorescence kinetics, solution NMR, BLI assay, AFM and EM techniques, which all have been carried out carefully and with state of the art expertise. This study is of prime interest in the field of amyloid aggregation and chaperone proteins.

I only have a few minor remarks:

(1) ThT fluorescence kinetics reveal that both human and archeal PFD exert inhibition effects at substoichiometric concentrations, acting mainly on secondary nucleation and elongation pathways. Although no major effect on lag phase is claimed (page 14), there seems to be an effect at high concentrations of PFD, suggesting that PFD might also affect the primary nucleation pathway. I suggest adding graphs showing the effect on the lag time in supplementary material.

(2) In the NMR interaction experiments, there is no control about the state of IAPP at the end of recorded experiments. Considering the concentration of IAPP used (200 μ M, much higher than in ThT assays), the temperature (30°C) and the acquisition times (0.5 day), is there any aggregation of IAPP under these conditions? Similarly what is the recording time of 15N-SOFAST-HMQC on 15N-IAPP samples?

(3) The docking models were calculated with HADDOCK using a set of interacting residues in IAPP and PhPFD inferred from NMR data obtained on monomeric IAPP (pages 11 and 28). If this strategy is sound to model the complex between PhPFD and monomeric IAPP, it is less clear how the PFD complexes to the fibril surface and to the fibril ends agree with the set of restraints (discussed page 15). An additional figure in Supplementary material would be helpful to show the interacting residues of IAPP on the fibril surface or on the fibril ends.

(4) The authors have used a recombinant form of IAPP for the NMR studies on 15N-labelled IAPP which differs from the native sequence by the absence of C-terminal amidation. Although this modification has probably minor effects, the authors should precise that the recombinant form is not amidated in the experimental section.

(5) Page 37, figure 3 legend: it is common practice to use a scaling factor between 15N and 1H chemical shift modifications to monitor CSPs. Is there any reason for not using standard CSP formula?

(6) The structural formula of IAPP-DOTA conjugate in figure 2a is not correct: there should be amide bonds between DOTA-betaAla and betaAla-IAPP.

Manuscript NCOMMS-21-40840

Revised version

Point-by-point response to the reviewers' comments

Colour code:

Black - Reviewers' remarks

Green - authors' comment

Blue - updated parts of the manuscript

Gray - non-modified parts of the manuscript

Reviewer #1 (Remarks to the Author):

In the manuscript by Törner and coworkers, the researchers subjected amylin fibrils to HSP60, the chaperonin present in the cytosol of archaeal and eukaryotic cells. The researchers determined interactions between the HSP60 and the fibrils using NMR and found, using electron microscopy, that HSPR 60 binds to the fibrils.

It seems that the reviewer did not realize that the protein of interest in this article is not the HSP60 Chaperonin (a hexadecameric foldase of almost 1 MDa) but its co-chaperonin, named Prefoldin (a hexameric holdase of less than 100 kDa). The authors regret that the title of the manuscript might have been misleading in its wording "Inhibition ... by the Hsp60 Co-Chaperonin Prefoldin" meaning inhibition by prefoldin which is the Hsp60 co-chaperonin. To avoid any possible confusion and to reach a better clarity, the title was changed to "Structural Basis for the Inhibition of IAPP Fibril Formation by the Co-Chaperonin Prefoldin".

(Page 1, title)

This work, nearly entirely repeats the studies published back in 2012 and 2013 by Robb and Lednev: doi: 10.1016/j.bbrc.2012.04.113 and doi.org/10.1021/cb400238a. The researchers didn't cite this work, which suggests that they very poorly reviewed the literature on the topic of their research.

In the light of the previously published work, there is very little if any novelty in the submitted manuscript.

The authors appreciate the suggested references and have cited in the introduction of the revised manuscript the mentioned studies in the context of general description of interaction between chaperones and amyloid fibrils, as the new references 29 and 30.

Nevertheless, the authors have to disagree on the point of absent novelty of the current study, since it is focused on the inhibition of disease-related IAPP aggregation by prefoldin, the co-chaperonin of Hsp60, which has not yet been reported for the time being. The authors think that such comment reflects a misunderstanding about the biological system at the centre of this article, and the comparison with suggested references is therefore not relevant in this context.

Also, although the structural part of the work is in the good shape, the authors fully overlooked the cell toxicity of the work; they neither evaluate the behavior aspect of their work on animal modes, which is expected for this highly ranked journal.

We thank the reviewer for this suggestion. In order to examine the cytotoxic effects of IAPP with or without (different) concentrations of PhPFD, the authors additionally performed a cell viability experiment (MTT test) on rat pancreatic beta cells (RIN-m5F). The test revealed that PhPFD is capable of reducing IAPP-induced toxicity. Acquired data were added to the manuscript as Fig. 1c, as well as an according paragraph – in the Results section.

Fig. 1c PhPFD reduces the IAPP-induced negative impact on cell viability of RIN-m5F cells. After incubation of RIN-m5F cells with 1 μ M IAPP, w/o or with several concentrations of PhPFD (0.1, 0.3, 1 μ M), a MTT cell viability test was conducted. Data revealed that co-incubation of equimolar concentrations of IAPP and PhPFD results in an abolition of IAPP-induced toxicity. Data is represented as mean \pm SD (out of two independent experiments with four to five technical replicates), one-way ANOVA with Tuckey post hoc analysis, ***: $p < 0.001$.

(Page 44, Fig. 1c; page 41, Legend of Fig. 1c)

“PFD increases the viability of rat pancreatic beta cells exposed to IAPP aggregates. To assess the effect of PFD on IAPP toxicity, we evaluated the viability of rat pancreatic RIN-m5F cells upon addition of IAPP with and without PFD in an MTT (3-(4,5-Dimethylthiazol-2-yl)-2,5-diphenyl-tetrazolium bromide) assay. When monomeric IAPP at a concentration of 51 μ M was incubated for 3 days and diluted into the cell culture medium to a final concentration of 1 μ M, cell viability reduced to 47.4% of that of the non-treated control (Fig. 1c). Co-incubation with PhPFD at molar ratios between 1:10 and 1:1 showed a concentration-dependent rescue of cell viability, with cell viability reaching 84.8% of the control at equimolar ratio of PFD. This demonstrates that PFD inhibits IAPP-induced cell toxicity (Fig. 1c).”

(Page 9, top)

Reviewer #2 (Remarks to the Author):

The manuscript entitled « structural basis for the inhibition of IAPP fibril formation by the Hsp60 co-chaperonin Prefoldin » examines the effects of HSP60 type II co-chaperonin prefoldin protein (PFD) on the human islet amyloid polypeptide (hIAPP) fibril formation and structural conformation. hIAPP is an amyloidogenic peptide that form fibril in the pancreas of type 2 diabetes patients. In this paper, the authors investigated the hIAPP fibril formation in the presence of PFD using ThT kinetics fluorescence and atomic force microscopy. They examined the hIAPP/PFD interactions using NMR and were able to elucidate the mechanisms of inhibition. This is clearly an interesting paper. The manuscript is well-written. I recommend publication of this paper after the following corrections.

1) My main concern is the images obtained with AFM. The authors stated that the samples are dried. Why did they dry the sample since we know that the drying step induce some changes in the morphologies of the samples? I recommend strongly the authors to perform again the AFM images without the drying step. This is also maybe the reason of the observation of the clustered aggregate of IAPP in the presence of PFD.

The authors appreciate reviewer's concern and recommendation, hence additionally performed the AFM imaging in solution. This resulted in a similar tendency, and the clustered IAPP aggregates in presence of human PFD were still detectable.

“**Supplementary Fig. 2c** Height (top) and phase (bottom) 5x5 μ m² AFM images performed in liquid confirming the tendencies reflected by the AFM images after a drying step (shown in (a) and (b)) and demonstrating that the clustered aggregates appear not due to sample drying, but they are present already in the solution.”

(Page 4 in Supplement, Fig. S2c; page 5 in Supplement, middle)

Although the authors were and are aware of the fact that the drying step induces some changes in the sample morphology, this way of sample preparation was chosen in order to be able to immobilize on the surface possibly all the species from the solution, independent of their affinities towards the substrate surface. These affinities might significantly differ between the IAPP fibrils alone and the aggregates formed in presence of PFD, causing uneven probabilities of such objects to be detected without being relatively harshly immobilized on the surface. For this reason, the authors prefer to consider the images

performed after the drying step to more reliably reflect the sample content. Nevertheless, the images performed in liquid were added as Supplementary Fig. S2c to clearly demonstrate that the clustered aggregates appear not due to sample drying, but they are present already in the solution.

2) Some kinetics analysis is not also well described and will be difficult to understand. For example, this is not obvious that the experiment without agitation and with preformed fibrils will give information on the secondary nucleation. Same remark for the experiment with the sonicated, short preformed fibrils on the elongation. I recommend the authors to re-write the part on the kinetics analysis and explain more the analysis and give more references. In addition, the terms “lag time corrected” in figure S1 should be explained.

The authors acknowledge the fact that the kinetic analysis description should be more specific, as well as the term “lag-time-corrected”, and added an according part in the Methods section:

“Experimental data shown in Fig.1 for *de novo* and secondary nucleation assays were lag-time-corrected before averaging over the triplicates: lag time for each replicate was set to the mean lag time of the according triplicate by a slight shift of unmodified curves along the time axes. Lag time, defined here as the time before the first order time derivative reaches 5% of its maximum value, varied in triplicates on average about 6%. An exemplary comparison of modified and non-modified data is shown in Supplementary Fig. S1b.

Extracting the kinetic parameters was done automated using a custom script, processing the data uniformly. First, data was smoothened via moving averaging over 11 data points. Then, numerical approximations of first and second order time derivatives were calculated as the symmetric difference quotient of the ThT FI data and first order derivative accordingly, using a derivative depth window of 4 points.

Growth phase duration was defined as time between growth start and end (time points where first derivative has a value of 5% of its maximum); maximal growth rate – as maximal value of first order derivative; average growth rate – as the difference between ThT FI values at end and start of growth, divided by the growth phase duration; initial growth rate was calculated as mean value of first 20 values of the first order derivative; final plateau height – as a mean of 50 points following after the growth end; acceleration maximum – as the maximal value of second order time derivative. The full text of the used script is linked to this publication.”

(From page 24, end to page 25, middle)

Regarding the point about addition of sonicated or non-sonicated fibrils, the following corrections and references were added to the manuscript:

“For seeded assays, pre-formed mature IAPP fibrils were added (8 to 9% of overall IAPP concentration in monomer equivalent): either non-sonicated in the secondary nucleation assay in order to provide the fibril surface capable of catalyzing nucleation³³, or sonicated (Sonopuls MS 72 microtip sonotrode, Bandelin; 10% amplitude, 4 cycles of 1 s pulse and 5 s break) in the elongation assay, thus providing a higher number of fibril ends to recruit and incorporate monomers resulting in the linear growth of fibril mass⁴⁴.”

(Page 23, middle)

Also, the part “PFD impedes IAPP aggregation” of the Results section was re-written and now includes more explicit description and references concerning secondary nucleation and elongation processes:

“We tested the effects of human and archaeal PFD on two specific steps of IAPP fibril formation, fibril surface-catalyzed secondary nucleation and fibril elongation, by performing aggregation assays under conditions where one of these steps is dominating the overall aggregation kinetics. Secondary nucleation has been identified as a critical step in IAPP fibril formation, and has been linked to the emergence of toxic IAPP aggregates^{18,40}. Under conditions with an active secondary nucleation pathway, addition of fibril seeds bypasses the need for primary nucleation⁴¹. We therefore tested the effect of PFD on IAPP assembly in the presence of preformed IAPP fibril seeds (Fig. 1a, middle row). In order to suppress aggregation pathways that are independent of secondary nucleation, : 1) we did not agitate the samples, as agitation promotes the amplification of aggregates formed by primary nucleation, by increasing the effective size of surfaces involved in primary nucleation, by enhancing mass transport, and by promoting mechanical fibril breakage⁴², and 2) we did not sonicate the preformed fibrils, which would promote aggregation by pure elongation of the seeds. Under these conditions of dominant secondary nucleation, maximal growth rate drastically decreases for [PFD]:[IAPP] molar ratios above 1:113, and drops to almost zero for molecular ratios larger than 1:17, indicating that PFD effectively interferes with the fibril surface-catalyzed generation of new fibril nuclei. Sub-stoichiometric PFD concentrations suffice, suggesting that fibril surfaces are the target sites of PFD in inhibition of secondary nucleation (see Arosio et al. for an overview of potential target species for inhibition of specific steps of amyloid fibril formation⁴³).

Fibril elongation, i.e. the binding of monomers to fibril ends and their conformational conversion into the fibrils’ cross- β structure, is dominating the aggregation kinetics when a high concentration of fibril ends is present⁴⁴. For observation of PFD’s effect on elongation of IAPP fibrils we therefore added sonicated, short, pre-formed IAPP fibrils to offer a high number of fibril ends (Fig. 1a, bottom row). Under these conditions, the very first phase of linear growth reflects the pure elongation process, while the subsequent exponential increase indicates that secondary nucleation starts to contribute. Addition of PFD results in a distinct decrease of the initial growth rate, demonstrating that PFD interferes with fibril elongation. Sub-stoichiometric PFD concentrations are sufficient, indicating that fibril ends are the target sites of PFD in inhibition of fibril elongation.

While the described effects on IAPP fibrillation are present for both archaeal and human PFD, the inhibitory effects of hPFD are more pronounced. The kinetic study of different modes of IAPP aggregation in presence of PFD shows that PhPFD and hPFD affect various steps of the fibrillation process. ~~Sub-stoichiometric inhibition of both elongation and secondary nucleation implies interaction of PFD with IAPP fibrils, at the fibril ends and also along the fibril surface.~~ Moreover, the final plateau values in all the kinetic experiments become significantly lowered if PFD is present.”
(From page 6, bottom to page 8, middle)

3) It seems that PFD is found in the cytosol of cells. However, hIAPP is matured in the secretory granules and secreted directly in the extracellular domain. Do you think that hIAPP and PFD could be in the same place in cells?

We thank the reviewer for this comment. We have included in the revised manuscript a supplementary paragraph to address this point:

“PFD increased the viability of IAPP-treated cultured rat pancreatic beta cells in an MTT assay (Fig. 1c). For this cytoprotective effect to prevail *in vivo*, PFD and IAPP would need to colocalize in a common cellular compartment. While PFD is localized in the cytosol, IAPP is processed from its prohormone in the Golgi and in the insulin secretory granule, from where it is secreted⁶⁰. Thus, freshly processed and secreted IAPP is unlikely to be a PFD client. However, IAPP oligomers might escape from the secretory pathway, leading to cytosolic IAPP aggregates^{60,61}. Moreover, extracellular IAPP oligomers have been shown to be taken up by beta cells, resulting in accumulation of cytoplasmic IAPP and altered cellular proteostasis^{62,63}. Such cytosolic IAPP species might well constitute PFD clients.”

(Page 17, end to page 18, top)

Moreover, we expanded the following sentence in the discussion: “Here we apply IAPP as a model amyloidogenic substrate and show that prefoldin inhibits IAPP fibrillation at sub-stoichiometric concentrations and interacts with multiple IAPP species, such as monomers and mature fibrils, and propose models for these interactions.”

(Page 14, top)

4) minor corrections:

Do the well plates used in this study are black or transparent? The author should add this information.

The concentration of hIAPP used in the study should be mentioned on the legend of the figures.

The authors fully agree on these points and modified the manuscript accordingly.

(Currently: “96-well half-area polystyrene non-binding surface (NBS) microplate, black with flat transparent bottom (3881, Corning)” in Methods, “5 μ M IAPP alone” in the figure legend and “IAPP (5 μ M) aggregation kinetics...” in figure description.)

(Page 23, end; page 44, Fig. 1a, top left; page 40, top)

Reviewer #3 (Remarks to the Author):

In this paper, R. Törner et al. provide for the first time a molecular basis to understand the mechanism of inhibition of IAPP amyloid fibril formation by prefoldins, which are chaperones belonging to the Hsp60 co-chaperonin family. The results are based on a comprehensive set of complementary techniques, namely ThT fluorescence kinetics, solution NMR, BLI assay, AFM and EM techniques, which all have been carried out carefully and with state of the art expertise. This study is of prime interest in the field of amyloid aggregation and chaperone proteins.

I only have a few minor remarks:

(1) ThT fluorescence kinetics reveal that both human and archeal PFD exert inhibition effects at substoichiometric concentrations, acting mainly on secondary nucleation and elongation pathways. Although no major effect on lag phase is claimed (page 14), there seems to be an effect at high concentrations of PFD, suggesting that PFD might also affect the primary nucleation pathway. I suggest adding graphs showing the effect on the lag time in supplementary material.

We thank the reviewer for pointing out the potential effect of PFD on primary nucleation. It is correct that there is a minor increase in lag time with increasing PFD concentrations, if the lag time is defined by achievement of a threshold of Thioflavin T fluorescence intensity. With the definition of lag time used by us (the time before the first order time derivative reaches 5% of its maximum value) this minor effect is not evident and even reverts at high PFD concentrations (see figure, now included in Supplementary Fig. S1a). Importantly, alterations in the lag time cannot be unambiguously assigned to effects on primary nucleation (see for example Fig. 2b, middle panel, in Arosio et al., <https://doi.org/10.1016/j.tips.2013.12.005>). Therefore, our *de novo* kinetics data does not allow us to draw conclusions on the precise effects of PFD on the different reaction steps under the applied *de novo* reaction conditions (absence of seeds, agitation). (Page 2 in Supplement)

(2) In the NMR interaction experiments, there is no control about the state of IAPP at the end of recorded experiments. Considering the concentration of IAPP used (200 μM , much higher than in ThT assays), the temperature (30°C) and the acquisition times (0.5 day), is there any aggregation of IAPP under these conditions? Similarly what is the recording time of 15N-SOFAST-HMQC on 15N-IAPP samples?

We thank the reviewer for this critical comment. The authors were aware of this point and have designed all NMR experiments in order to include internal control to ensure that IAPP state did not change during the longest experiment as explained below:

- For titration of U-[^{15}N]-IAPP with PFD, the IAPP concentration was 29 μM . The length of the 2D experiments (^{15}N -SOFAST-HMQC) depended on the signal-to-noise ratio, as strong signal decrease was observed upon addition of PFD. Therefore, the longest experiment was acquired for an IAPP:PFD ratio of 1:8. Split acquisition schemes were implemented for experiments longer than 5 h to control that the IAPP signals were stable during the course of the experiment. For example, the ^{15}N -SOFAST-HMQC experiment for the ratio 1:8 was split into 3 experiments, each one with a duration of 5 h 35 min. The panel (a) below displays the projection in ^1H dimension of IAPP signals for each one of the three experiments. The three spectra are superimposable. The following sentence has been added in Methods to clarify the point: “For 2D titration experiments with an acquisition time longer than 5 hours, the NMR data acquisitions were split to control that the IAPP NMR signals did not decrease over time, ensuring that the IAPP state did not change over the course of the NMR experiment.”

(Page 29, top)

- For PRE measurement, the longest experiments were split into shorter experiments and 1D control experiments were inserted to verify that the intensity of NMR signals of IAPP were stable. The longest experiment was acquired with IAPP- Gd^{3+} for PRE measurements on PFD NH moieties. 3 individual experiments were acquired with a duration of 4 h 40 min for each one. 1D spectra acquired before and after each experiment are presented below on panel b). The spectra are presenting methyl signals of IAPP- Gd^{3+} . These spectra are superimposable with each other. The experiments were acquired on a sample of 100 μM [^{15}N , ^2H]-PFD in presence of 100 μM IAPP- Gd^{3+} . All other PRE experiments were acquired for shorter experimental time. This is now specified in Methods section: “For long PRE experiments, the NMR data acquisitions were split and 1D control NMR experiments were inserted before and after each 2D experiment. These control experiments were enabling to monitor IAPP NMR signals over time, ensuring that the IAPP state did not change over the course of the NMR experiment.”

(Page 30, top)

- Note that in the initial version, one sentence in Methods about the IAPP concentration for PRE experiments was confusing. We have modified this sentence to indicate that the maximum concentration of IAPP was 100 μM : “For [^{15}N , ^2H]-PhPFD the resulting concentrations were 100 μM PFD and 100 μM IAPP, for $^{13}\text{CH}_3$ -labelled U-[^2H]-PhPFD ~50 μM PFD and 100 μM IAPP.”

(Page 29, end)

These control experiments clearly established that the IAPP did not precipitate nor fibrillate during the NMR experiments. The fact that IAPP aggregation in NMR experiments starts at much higher concentration compared to the ThT assays can be attributed to the lack of air-water interface (AWI) in an NMR tube, since AWI is known to play an important role in amyloid aggregate formation (e.g. Campioni S. *et al.* The Presence of an Air–Water Interface Affects Formation and Elongation of α -Synuclein Fibrils. *J. Am. Chem. Soc.* **2014** *136* (7), 2866-2875; DOI: 10.1021/ja412105t, or much earlier Schladitz *et al.* Amyloid-beta-sheet formation at the air-water interface. *Biophys J.* **1999** Dec; *77*(6):3305-10. DOI: 10.1016/S0006-3495(99)77161-4)

(3) The docking models were calculated with HADDOCK using a set of interacting residues in IAPP and PhPFD inferred from NMR data obtained on monomeric IAPP (pages 11 and 28). If this strategy is sound to model the complex between PhPFD and monomeric IAPP, it is less clear how the PFD complexes to the fibril surface and to the fibril ends agree with the set of restraints (discussed page 15). An additional figure in Supplementary material would be helpful to show the interacting residues of IAPP on the fibril surface or on the fibril ends.

Following the suggestion of the reviewer, we have added a supplementary figure (Supplementary Fig. S11) presenting a) fibril surface with corresponding amino acids used as docking restraints coloured in red and yellow, b) and c) same surfaces with the IAPP residues in contact with PFD residues ($d < 6 \text{ \AA}$) coloured in green on the surface of the docking models. Panel a) indicates that despite both the N-terminal and middle interaction regions are defined as restraints, only amino acids 6-19 are accessible on the fibril surface to form a complex with PFD, while amino acids of the middle interaction region are buried in the fibril core. The amino acids of the middle interaction region are partially accessible only on fibril ends. We observed a good correlation between amino acids defined as docking restraints and interacting residues observed in either fibril surface or fibril end docking models.

Supplementary Fig. S11 Representation of residues of IAPP on the fibril surface or on the fibril ends interacting with PFD. **a** Amino acids used as docking restraints (see Methods section) are colour-coded in red (N-terminal binding segment) and yellow (middle interacting segment). Panel **(b)** presents in green the amino acids of the IAPP fibril surface/PFD docking model that are distant by less than 6 Å from PFD residues. **c** Same as **(b)** with fibril end/PFD docking model. (Pages 14 in Supplement)

(4) The authors have used a recombinant form of IAPP for the NMR studies on ^{15}N -labelled IAPP which differs from the native sequence by the absence of C-terminal amidation. Although this modification has probably minor effects, the authors should precise that the recombinant form is not amidated in the experimental section.

The authors appreciate this remark and fully agree, thus added the according statement in “Protein preparation: IAPP” part of the manuscript:

“U- ^{15}N - or U- ^{15}N , ^{13}C] isotopically labelled human IAPP was recombinantly expressed in *Escherichia coli* in M9 medium supplemented with either 2 g/l ^{15}N - NH_4Cl and 3.2 g/l unlabelled glucose or with 2 g/l ^{15}N - NH_4Cl and U- ^{13}C] glucose and purified according to established protocols⁶⁴, resulting in production of IAPP in its non-amidated form.”

(Page 20, end)

(5) Page 37, figure 3 legend: it is common practice to use a scaling factor between ^{15}N and ^1H chemical shift modifications to monitor CSPs. Is there any reason for not using standard CSP formula?

The authors appreciate this remark and have modified the figure using a scaling factor of 0.14 between ^{15}N and ^1H chemical shift (<https://doi.org/10.1016/j.pnmrs.2013.02.001>). As it can be seen on comparison below, there are no major changes in the results obtained with or without a scaling factor of 0.14 for nitrogen shifts. The new figure in the revised version now includes the scaling factor.

(Page 46, Fig. 3b; page 42, top)

(6) The structural formula of IAPP-DOTA conjugate in figure 2a is not correct: there should be amide bonds between DOTA-betaAla and betaAla-IAPP.

The authors thank the reviewer for drawing attention to this matter and corrected the Fig. 2a in the revised version.
(Page 45, Fig. 2a)

REVIEWERS' COMMENTS

Reviewer #1 (Remarks to the Author):

The authors addressed all raised concerns and I suggest to accept the manuscript in the current form.

Reviewer #2 (Remarks to the Author):

The authors have improved the manuscript and answered my queries. The manuscript can be published.

Reviewer #3 (Remarks to the Author):

The authors have taken into account the different raised points and have carefully addressed these points in the revised version. Therefore I recommend publication of this manuscript.